# LeGIT: LLM Guided Intervention Targeting for Online Causal Discovery

## Abstract

A fundamental challenge in online causal discovery is designing effective experiments by selecting optimal intervention targets. Conventional numerical methods struggle in the early stages when limited interventional data is available, often yielding noisy or misleading selection guidance. In this work, we introduce the **L**arge **L**anguag**e** Model **G**uided **I**ntervention **T**argeting (LeGIT), a novel collaborative framework that synergizes the vast world knowledge of LLMs with the precision of numerical algorithms. By analyzing the meta-information of the causal system, it proposes highly informative intervention targets, effectively bootstrapping the discovery process to augment existing numerical approaches, while retaining the convergence guarantees. Evaluated across four realistic benchmarks, LeGIT demonstrates significant improvements in performance and robustness over existing methods, and even surpasses humans. This work establishes that LLMs can play a pivotal role in experimental design, offering a scalable and cost-efficient strategy to accelerate causal and scientific discovery.

## 1 Introduction

Science originates along with discovering new causal knowledge with *interventional experiments inspired by observations* (Kuhn & Hawkins, 1963). The art of finding causal relations from different interventions is then summarized and improved with statistical methods (Pearl & Mackenzie, 2018; Spirtes et al., 2010; Glymour et al., 2019). Identifying and utilizing causal relations is fundamental to numerous applications, including biology (Vowels et al., 2022) and financial systems (Dong et al., 2023). Despite the wide deployment of causal discovery methods, uncovering the underlying causal connections merely based on observational data alone is typically challenging due to limitations in identifiability. Mitigating this limitation usually requires additional interventional data obtained by perturbing part of the causal system to overcome the limited identifiability issue (Spirtes et al., 2000).

However, collecting interventional data is expensive and time-consuming, as it usually involves a physical process of a real-world system (Cherry & Daley, 2012). Consequently, *both the number of samples and the intervention targets are significantly limited in the experimental design in the real world* (Tong & Koller, 2001). Previous approaches usually rely on uncertainty (Lindley, 1956) or information theoretic metric to maximize the utility of an experiment (Tigas et al., 2022). Recently, leveraging gradient signals for intervention targeting has gained significant success (Olko et al., 2023), as it naturally fits into various gradient-based causal discovery methods. Despite some success, both uncertainty-based and gradient-based approaches may still suffer from suboptimality, as the estimation of the signals is usually noisy. Especially when with limited interventional data, the inaccurate estimation of the scores can easily mislead the intervention targeting and the subsequent causal discovery. The emergence of large language models (LLMs) (OpenAI, 2023) provides an opportunity to incorporate extensive world knowledge about experimental design into the intervention targeting process. It therefore raises an intriguing research question:

> *Can we leverage LLMs for intervention targeting and do LLMs really help with it?*

Recent explorations into the use of LLMs for various causal learning and reasoning tasks suggest that these models may already encapsulate substantial domain knowledge (Kiciman et al., 2023; Lampinen et al., 2023; Abdulaal et al., 2024; Li et al., 2024). LLMs have demonstrated the ability to process the meta-information encoded in natural language and leverage the meta-information to reason for

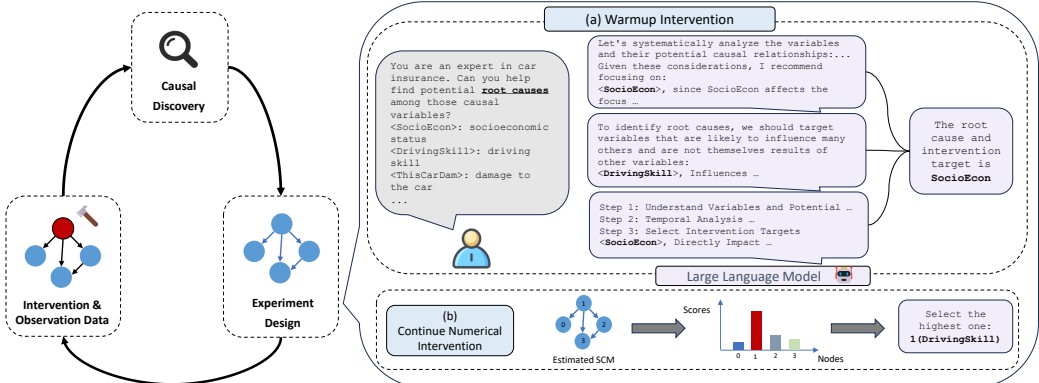

Figure 1: Illustration of the LeGIT framework. The left side represents the loop of Online Causal Discovery, while the right side illustrates the experiment design process. In Step (a), Large Language Models (LLMs) warm up the causal discovery process by leveraging world knowledge and aligning it with the experiment's meta-information. This enables the identification of clear causal structures, which, in Step (b), guide previous methods to pinpoint informative intervention targets effectively.

causality, which was considered restricted to humans (Gopnik et al., 2004; Trott et al., 2022; Sahu et al., 2022). Furthermore, LLMs have exhibited remarkable potential in advancing complex scientific discovery (AI4Science & Quantum, 2023). Additionally, discussions about the limitations of LLMs in understanding causality were also raised in the community (Zečević et al., 2023; Jin et al., 2023; Zhang et al., 2023a). This underscores the need for a robust approach that optimally extracts the world knowledge embedded in LLMs about experimental design while mitigating the risks of being misled by their hallucinations regarding causality (Zhang et al., 2023b).

To this end, we present a new framework called **L**arge Languag**e** Model **G**uided **I**ntervention **T**argeting (LeGIT), designed to maximize while robustly leveraging the knowledge in LLMs to assist with the intervention targeting. Shown as in Fig. 1, at the beginning of the causal discovery, the numerical-based methods have limited numerical knowledge about the underlying causal system to use due to the limited data. Consequently, the estimated signals tend to be noisy and misleading. In contrast, LLMs can leverage the meta-information about the causal system and relate the learned world knowledge to identify high-potential intervening targets. After obtaining a relatively clearer causal graph, LLMs may not be able to provide sufficient guidance. Therefore, similar to humans, LeGIT leverages numerical methods to select the intervening targets. Our contributions can be summarized as follows:

- To the best of our knowledge, we are the first to investigate the use of LLMs in the experimental design to select intervention targets for causal discovery.
- We propose a novel framework called LeGIT that combines the advantages of both the previous numerical methods as well as the LLMs to facilitate the intervening targeting.
- We conduct extensive experiments with 4 real-world based benchmarks and verify that LeGIT can outperform previous numerical-based methods and even humans.
- We highlight the promise of LLMs in causal and scientific discovery, that LLMs can effectively incorporate world knowledge, making them valuable cost-efficient complements to humans.

**On the data contamination of LLMs for online causal discovery.** A critical consideration when employing Large Language Models for scientific tasks on established benchmarks is the potential for training data contamination, as the benchmarks used in this work are from classic Bayesian network datasets (Scutari, 2010). Nevertheless, Jiralerspong et al. (2024); Khatibi et al. (2024); Long et al. (2023b) show that prominent LLMs struggle to accurately reconstruct the causal graphs of these very benchmarks from node descriptions alone. In addition, there is limited knowledge existing in the Internet on designing experiments for the Bayesian network datasets (Scutari, 2010). Hence, merely using LLMs to select the intervention targets is insufficient for online causal discovery, as also verified in our experiments.

## 2 PRELIMINARIES

We begin by briefly introducing the preliminaries and notation in online causal discovery (Olko et al., 2023).

Causal relations among variables can be modeled using Structural Causal Models (SCMs) (Pearl & Mackenzie, 2018; Glymour et al., 2019), where each variable $X_i$ is generated by $X_i = f_i(PA_i, U_i)$, with $PA_i$ its causal parents and $U_i$ independent noise. These relations can be represented by a directed acyclic graph (DAG) $G = (V, E)$, where nodes correspond to variables and edges represent direct causal links. The joint distribution factorizes as $P(X_1, ..., X_n) = \prod_{i=1}^{n} P(X_i|PA_i)$. However, observational data alone can only identify the DAG up to a Markov Equivalence Class (MEC) (Spirtes et al., 2000).

---

**Algorithm 1** ONLINE CAUSAL DISCOVERY

**Require:** Causal discovery algorithm $\mathcal{A}$ (e.g., ENCO), Number of data acquisition rounds $T$, Intervention targeting method $\mathcal{M}$, Observational dataset $\mathcal{D}_{obs}$
**Output:** Final parameters of graph model: $\varphi_T$ and Final estimated CausalDAG: $\mathbb{P}(G)$
1: $\mathcal{D}_{int} \leftarrow \varnothing$
2: Fit graph model $\varphi_0$ with algorithm $\mathcal{A}$ on $\mathcal{D}_{obs}$
3: **for** each intervention acquisition round $i = 1, 2, \ldots, T$ **do**
4:   $I_i \leftarrow$ generate intervention targets using $\mathcal{M}$
5:   $\mathcal{D}_{int}^i \leftarrow$ query for data from interventions $I_i$
6:   $\mathcal{D}_{int} \leftarrow \mathcal{D}_{int} \cup \mathcal{D}_{int}^i$
7:   Fit $\varphi_i$ with algorithm $\mathcal{A}$ on $\mathcal{D}_{int}$ and $\mathcal{D}_{obs}$
8: **end for**

---

To recover the true DAG from the MEC, online causal discovery incorporates interventional data (Tong & Koller, 2001; Hauser & Bühlmann, 2011; Ke et al., 2019). As outlined in Algorithm 1, a causal discovery algorithm $\mathcal{A}$ iteratively updates its structure using both observational and interventional data. Interventions, modeled as replacing $P(X_i|PA_i)$ with $\widehat{P}(X_i|PA_i)$, yield modified distributions $P_i(X) = \widehat{P}(X_i|PA_i) \prod_{j \neq i} P(X_j|PA_j)$. We use hard interventions for simplicity. The online discovery proceeds in $T$ rounds: Initially, a causal graph model $\phi_0$ is fitted using observational data. In each subsequent round, an intervention target $I$ is selected using a targeting method, and new interventional samples are collected to update the DAG.

Intervention targeting methods include Active Intervention Targeting (AIT), which uses an $F$-test (Scherrer et al., 2021), and Bayesian Optimal Experimental Design, which selects targets via posterior inference over DAGs (Tigas et al., 2022). Gradient-based Intervention Targeting (GIT) (Olko et al., 2023) instead leverages gradient signals from gradient-based causal discovery to estimate the utility of each intervention target via hallucinated gradients (Ash et al., 2020), offering improved performance and natural integration with gradient-based methods such as ENCO (Lippe et al., 2022), which we focus on in this study.

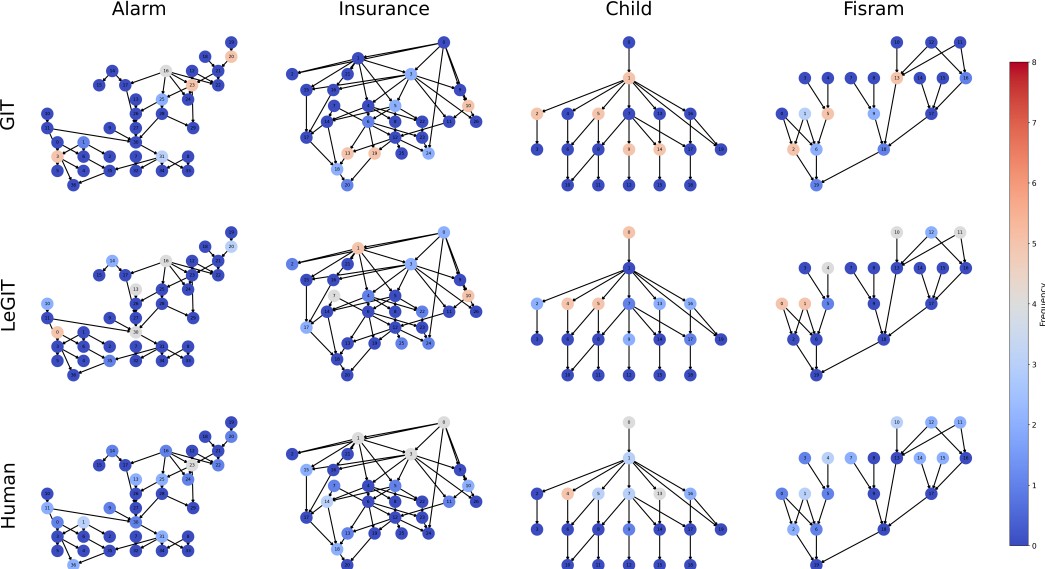

Figure 2: The selected Node Frequency obtained by different strategies on the initial stage from 5 different seeds. Frequency refers to the number of times a node is selected.

## 3 METHODOLOGY

### 3.1 CHALLENGES IN EXISTING INTERVENTION TARGETING

Despite the success of the GIT method, similar to other estimation-based approaches, GIT is highly sensitive to the accuracy of the gradient estimation and estimated causal graphs, which can be extremely noisy in the early rounds of an experiment. Therefore, we might mistakenly choose a variable that exerts minimal influence on the system, wasting valuable intervention budgets and misdirecting subsequent learning steps.

To demonstrate the above issue and the challenges in the existing intervention targeting methods more concretely, we consider four realistic causal discovery BN benchmarks (Scutari, 2010), i.e., *Alarm, Insurance, Child, and Fisram*, and plot the distribution of the intervention target at the initial stage.

As given in Fig. 2, it can be found that the success of GIT varies across different datasets. Intuitively, at the beginning of the intervention, intervening on variables that affect lots of other variables can bring more information about the system (Lindley, 1956; Agrawal et al., 2019). In the Alarm dataset, the selected intervention targets are influential nodes. However, in the Insurance, Child, and Fisram dataset, the selected nodes only influence a few other nodes. Intervening on such targets with limited influence may lead to resource waste and further misdirect subsequent online causal discovery rounds.

> You are a helpful assistant and expert in Car Insurance system research. Assuming we can do interventions to all the variables, your job is to assist in designing the best intervention experiments among the following variables to help discover variables' causal relations:
>
> <variable name>: Variable descriptions
>
> Assuming we can do interventions to all the variables, given the aforementioned variables and their descriptions, can you echo your knowledge about those variables, temporally analyze their relations, and then choose the best 5 intervention targets from all the variables, which hopefully are the root causes of the other variables to start our analysis of their causal relations?
> Let's think and analyze step by step. Then, provide your final answer (variable names only) within the tags <Answer>..</Answer>, separated by ",

Figure 3: Prompt template at warmup stage.

In comparison, we construct prompts to inquire LLMs about the potential root causes in the system, given only the meta-information, such as the variable descriptions (see Fig. 3). To investigate the effectiveness of LLM intervention targeting, we visualize the suggested intervening targets by LLMs in Fig. 2. It can be found that given only the meta-information, LLMs are able to relate the rich world knowledge to locate the desired influential nodes[1]. For example, in the Insurance dataset, LeGIT identifies SocioEcon (socioeconomic status, node 1) as a crucial factor, plausibly influencing car ownership, driving behavior, and access to safety features, while GIT cannot.

### 3.2 LARGE LANGUAGE MODEL GUIDED INTERVENTION TARGETING

Motivated by the aforementioned experiments, we present our framework **L**arge **L**anguag**e** Model **G**uided **I**ntervention **T**argeting (LeGIT) to combine the strengths of both numerical-based methods and LLMs to facilitate the intervention targeting. The description of the algorithm of LeGIT is given in Algorithm 2. LeGIT consists of four stages.

**Warmup Stage.** Since at the very beginning of the online causal discovery, numerical-based estimations are noisy and easily mislead the online causal discovery, we begin by prompting LLMs to relate the pre-trained knowledge, analyze the variable description, and suggest influential candidates. The prompt template is given in Fig. 3. The prompting will give the beginning list of intervention targets $\mathcal{D}_{\text{warmup}}$. From $\mathcal{D}_{\text{warmup}}$, we will select $T_{\text{warmup}}$ variables to obtain a basic map of the underlying causal system. For a robust performance, we perform self-consistency prompt skill (Wang et al., 2022) to get the final targets for a robust performance.

**Bootstrapped Stage.** Although the first warmup stage yields a basic structure of the underlying causal system, due to the intrinsic limitations of LLMs such as limited context length (Liu et al., 2023) and hallucination (Zhang et al., 2023b), LLMs may only focus on a subset of the variables and find the influential nodes therein. Nevertheless, when the number of causal variables is large, LLMs tend to give an incomplete set of influential nodes. Therefore, we further incorporate a second warmup stage to bootstrap the use of LLMs' world knowledge in early intervention targeting.

---

[1]We provide the summary of the number of neighbors for each node in Appendix B Fig. 11-Fig. 14.

---

**Algorithm 2** LeGIT: LARGE LANGUAGE MODEL GUIDED INTERVENTION TARGETING

---
**Require:** Causal discovery algorithm for Intervention Data $\mathcal{A}$ (e.g., ENCO); Intervention Score targeting method $\mathcal{M}_I$ (e.g GIT); LLM for root cause proposal $\Psi$; Number of data acquisition rounds $T$; Observational dataset $\mathcal{D}_{obs}$; Graph Node List $V$; Warmup Epoch $T_{\text{warmup}}$; Bootstrapped Search Epoch $T_{\text{bootstrapped}}$

**Ensure:** Final parameters of graph model: $\varphi_T$ and CausalDAG: $\mathbb{P}(G)$

1: $\mathcal{D}_{\text{warmup}} \leftarrow \Psi(V, T_{\text{warmup}})$ //Get Warmup List from LLM
2: **for** round $i = 1, 2, \ldots, T$ **do**
3:  **if** $i \leq T_{\text{warmup}}$ **then**
4:    $D_{int}^I \leftarrow \mathcal{D}_{\text{warmup}}[i]$
5:  **else if** $i = T_{\text{warmup}} + 1$ **then**
6:    // Get the Unvisited Nodes List
     $V_{\text{unvisited}} \leftarrow$ Unvisited nodes from $\mathbb{P}(G_i)$
7:    //Get Bootstrapped warmup Intervention Target from Unvisited Nodes
     $\mathcal{D}_{\text{bootstrapped}} \leftarrow \Psi(V_{\text{unvisited}}, T_{\text{bootstrapped}})$
8:    $D_{int}^I \leftarrow \mathcal{D}_{\text{bootstrapped}}[i - T_{\text{bootstrapped}}]$
9:  **else if** $T_{\text{warmup}} < i \leq T_{\text{warmup}} + T_{\text{bootstrapped}}$ **then**
10:    $D_{int}^I \leftarrow \mathcal{D}_{\text{bootstrapped}}[i - T_{\text{warmup}}]$
11:  **else if** $T_{\text{warmup}} + T_{\text{bootstrapped}} < i \leq 2(T_{\text{warmup}} + T_{\text{bootstrapped}})$ **then**
12:    //Re-sampling LLM's List
     $D_{int}^I \leftarrow (D_{\text{warmup}} + D_{\text{bootstrapped}})[i - T_{\text{warmup}} - T_{\text{bootstrapped}}]$
13:  **else**
14:    $D_{int}^I \leftarrow$ generate intervention targets using $\mathcal{M}_I$
15:  **end if**
16:  $\mathcal{D}_{int} \leftarrow \mathcal{D}_{int} \cup \mathcal{D}_{int}^I$
17:  Fit $\varphi_i$ with algorithm $\mathcal{A}$ on $\mathcal{D}_{int}$ and $\mathcal{D}_{obs}$
18: **end for**

---

More concretely, we leverage the intermediate causal discovery results $\varphi_{T_{\text{warmup}}}$ after the $T_{\text{warmup}}$ rounds and examine the left variables that have not been involved in $\mathcal{D}_{\text{warmup}}$. Then, we further prompt LLMs to give more focus on the left set of variables and to find the influential variables that were missing in previous rounds.

**Re-sampling Stage.** After getting the warmup and bootstrapped intervention target, we perform re-sampling to refine the intervention selection further, thereby improving the final accuracy of the algorithm while minimizing unnecessary interventions (Lippe et al., 2022). We interleave the two lists so that each proposed target must "survive" two independent votes before being used. This reduces the chance that a single hallucinated LLM suggestion dominates, and guarantees coverage of both influential and previously isolated nodes.

**Continual Intervention Stage.** After the three warmup stages, we have already obtained relatively clearer yet complicated causal graphs. Even for humans, it is hard to determine the best experimental design. Therefore, we switch to using the numerical-based methods to continue to consume the remaining intervention budgets.

### 3.3 THEORETICAL AND PRACTICAL DISCUSSION

After setting up the LeGIT algorithm, we discuss the convergence of LeGIT. Since LeGIT ends up with a numerical-based method for concluding online causal discovery, it follows intuitively that, like other numerical-based methods (e.g., GIT (Olko et al., 2023)), and an effective causal discovery algorithm, such as ENCO (Lippe et al., 2022), LeGIT can converge, further details available in Appendix D. Nevertheless, we empirically observe that LeGIT can converge to a better solution compared to the same numerical-based method without LLMs involved.

Consistent with prior work, we mainly adopt GIT as the numerical-based method $\mathcal{M}_I$ and ENCO as the gradient-based causal discovery method. However, as also suggested in GIT, ENCO can also be switched to other gradient-based methods. Additionally, LeGIT is also compatible with other numerical-based approaches.

## 4   RELATED WORK

**Intervention Targeting/Experiment Design.** Scientific progress in causal discovery is often driven by interventional experiments inspired by observational insights (Kuhn & Hawkins, 1963). Traditional methods focused on designing effective experiments to establish causal links, while statistical approaches aimed to automate causal inference from observational data (Pearl & Mackenzie, 2018; Spirtes et al., 2000). However, observational data alone is insufficient for identifying causal structures, and interventional data is costly to collect (Spirtes et al., 2000). To address these challenges, several methods for optimal intervention design have been developed.

AIT selects intervention targets using an $F$-test inspired criterion, evaluating discrepancies in interventional sample distributions from a posterior distribution of graphs (Scherrer et al., 2021). Causal Bayesian Experimental Design (CBED) uses Bayesian Optimal Experimental Design to select interventions that maximize mutual information (MI) between new data and existing graph beliefs, with MI estimated via a BALD-like method (Tigas et al., 2022; Houlsby et al., 2011). GIT (Olko et al., 2023) leverages gradient information to determine interventions that maximize impact on causal parameter updates, which is particularly advantageous in low-data settings. In our work, we explore leveraging these advanced intervention strategies within the framework of LLMs to determine whether LLMs can effectively engage in experimental design for causal discovery, pushing the boundaries of what automated, data-driven causal inference can achieve.

**Causal Discovery with LLM.** Recent advancements in LLMs have opened new opportunities in causal learning and reasoning by incorporating domain knowledge, common sense, and contextual reasoning (Kiciman et al., 2023). LLMs have demonstrated capabilities across Pearl's ladder of causation—association, intervention, and counterfactuals—bridging gaps that traditional models have with high-level causal reasoning. They have shown promising results in pairwise causal discovery tasks by utilizing semantic information not accessible through numerical data alone (Jiralerspong et al., 2024; Vashishtha et al., 2025).

On the other hand, LLMs can sometimes behave like "causal parrots", repeating learned associations without demonstrating true causal reasoning (Zečević et al., 2023; Chen et al., 2024). Moreover, their performance varies significantly depending on task complexity, with limited success in advanced causal reasoning such as full graph discovery and counterfactual analysis (Zhang et al., 2023a; Jin et al., 2023; Long et al., 2023a). Another promising line of work integrates LLMs with traditional causal discovery methods to leverage their complementary strengths (Long et al., 2023a; Abdulaal et al., 2024; Vashishtha et al., 2023; Liu et al., 2024). This hybrid approach has shown improved performance in constructing causal graphs, benefiting from LLMs' understanding of language context and traditional methods' data-driven precision.

While prior studies emphasize the role of LLMs in causal analysis, the question of whether LLMs can meaningfully contribute to experimental design in causal discovery remains largely unaddressed. Experimental design encompasses proposing interventions, predicting outcomes, and assessing experimental strategies—tasks that extend beyond basic causal inference. This paper seeks to bridge this gap by investigating the potential of LLMs to support experimental design, exploring their unique value, and critically evaluating their strengths and limitations in guiding causal experiments.

## 5   EXPERIMENTS

In this section, we conduct extensive experiments to evaluate LeGIT on real-world datasets and compare LeGIT against various baselines in intervention selection and humans. We provide a brief overview of the experimental setups here, with further details available in Appendix C.

### 5.1   EXPERIMENTAL SETUP

**Datasets.** Specifically, we use four real-world based benchmark datasets along with their corresponding ground truth causal graphs from the BN repository (BNMA; Scutari, 2010): *Fisram*, *Child*, *Insurance*, and *Alarm*. It provides causal graphs derived from real-world applications that are widely recognized as benchmarks. These datasets encompass a diverse set of professional scenarios, ranging from car insurance, ecosystem to medical systems, which are crucial to enhancing the knowledge captured by large language models (LLMs). More details are given in Appendix B.

Table 1: Average SHD and SID with standard deviation (over 5 seeds) for real-world data ($T = 33$ rounds, $|D_{int}^I| = 32$, $N = 1056$).

| Methods | Alarm (37 Nodes, 46 Edges) | | Insurance (27 Nodes, 52 Edges) | | Child (20 Nodes, 25 Edges) | | Fisram (20 Nodes, 23 Edges) | |
|---|---|---|---|---|---|---|---|---|
| | SHD↓ | SID↓ | SHD↓ | SID↓ | SHD↓ | SID↓ | SHD↓ | SID↓ |
| CBED | 28.20 ±4.31 | 213.80 ±42.44 | 21.60 ±4.63 | 260.00 ±31.83 | 5.40 ±2.06 | 44.40 ±18.51 | 3.60 ±1.62 | 27.20 ±6.31 |
| AIT | 32.80 ±8.42 | 204.60 ±52.09 | 24.20 ±7.47 | 312.40 ±87.50 | 9.00 ±3.29 | 52.20 ±21.03 | 8.00 ±3.63 | 53.00 ±29.23 |
| Random Choice | 38.80 ±3.54 | 204.40 ±58.15 | 26.00 ±3.63 | 323.80 ±14.96 | 5.40 ±1.20 | 51.00 ±17.11 | 4.40 ±3.38 | 41.00 ±37.38 |
| Round Robin | 25.00 ±1.26 | **118.60 ±21.78** | 17.40 ±4.54 | 232.20 ±27.23 | 3.40 ±2.50 | 23.00 ±14.39 | 4.80 ±1.72 | 57.20 ±19.61 |
| Degree Prob | 29.40 ±4.67 | 144.60 ±49.77 | 25.80 ±2.93 | 305.20 ±17.45 | 6.20 ±2.48 | 36.20 ±16.35 | 6.60 ±1.20 | 40.60 ±9.73 |
| GIT | 19.60 ±3.77 | 131.40 ±47.66 | 16.40 ±3.14 | 243.80 ±28.72 | 2.80 ±0.75 | 20.40 ±12.50 | 2.00 ±1.67 | 27.00 ±21.94 |
| **Human** | 22.60 ±5.43 | 133.20 ±27.01 | 14.20 ±3.43 | 232.20 ±40.74 | **2.00 ±0.63** | **18.80 ±8.42** | 1.60 ±1.02 | 21.20 ±12.61 |
| **LeGIT** | **17.40 ±3.61** | **121.00 ±38.27** | **12.60 ±0.80** | **200.60 ±35.32** | 2.20 ±0.98 | 20.60 ±5.61 | **1.20 ±0.98** | **15.80 ±8.23** |

**Baselines.** We compare **LeGIT** against different online causal discovery algorithms **GIT** (Olko et al., 2023), **AIT** (Scherrer et al., 2021), **CBED** (Tigas et al., 2022) as selection strategies for online active learning interventions, as well as four additional baselines following different heuristics:

1. **Random Choice**: A target node is randomly select from the set of all nodes at each step.

2. **Round Robin**: A target node is chosen randomly from the unvisited nodes at each step. Once all nodes are selected, the visitation counts are reset.

3. **Degree Prob Sample**: A target node is randomly chosen from all nodes, with selection probability normalized by each node's out-degree.

4. **Human**: We ask five *master's/Ph.D.-level* individuals, presenting them with the same information and process as provided to the LLMs.

Among the baselines, Degree Prob Sample can be considered as an *oracle* to LLM that adopts the out-degree of each node in the ground truth DAG. In addition, we also include the human baseline to better isolate and understand the unique contributions of LLMs.

**Implementation.** We employ the GPT-4O[2] (OpenAI, 2024) with **ENCO** (Lippe et al., 2022) as the backbone causal discovery algorithm, with detailed settings provided in the Appendix C.1. The observational dataset consists of $|\mathcal{D}_{obs}| = 5000$ samples, and we conduct $T = 33$ rounds of intervention sampling, with each round acquiring an interventional batch of $|\mathcal{D}_{int}^I| = 32$ samples, leading to a total of $N = 1056$ interventional samples. For GIT and AIT, we use $|\mathcal{G}| = 50$ graphs, each with $|\mathcal{D}_{G,i}| = 128$ data samples for the Monte Carlo approximation of the score. We set $T_{\text{warmup}} = 3$ and $T_{\text{bootstrapped}} = 2$ for LeGIT.

**Metrics.** We evaluate the performance of different methods using three metrics following the common practice: Structural Hamming Distance (SHD) (Tsamardinos et al., 2006), Structural Intervention Distance (SID) (Peters & Bühlmann, 2015), and Balanced Scoring Function (BSF) (Constantinou, 2019). SHD (lower is better) quantifies the number of edge insertions, deletions, or reversals needed to transform one graph into another. SID (lower is better) assesses causal inference by evaluating the correctness of the intervention distribution. BSF (higher is better) mitigates bias by balancing the evaluation of edges and independencies within Bayesian Network structures. A detailed description can be found in the Appendix C.2.

## 5.2 EMPIRICAL RESULTS

**Recovery of causal graph.** As shown in Table 1, it can be found that LeGIT achieves state-of-the-art causal discovery performances, with consistent improvements against the adopted gradient-based methods and *even human baseline* across all metrics and benchmarks. The superior SHD scores demonstrate that LeGIT is highly effective in accurately reconstructing the underlying graph structures, minimizing the number of erroneous edge modifications required.

**Intervention dynamics.** In Fig. 4, we further plot the performances of different methods along with the increase of the data samples obtained from different rounds. It can be found that, although at the beginning of the online causal discovery, LeGIT may not demonstrate outstanding SHD results. Along with more data samples coming in, LeGIT converge to a better solution faster than any other

---

[2]We used gpt-4o-2024-08-06 from the Azure platform. We also tested open-source models in Appendix C.3.

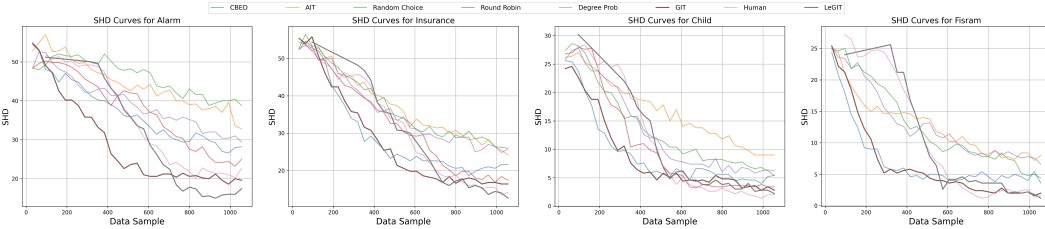

Figure 4: SHD metric for different methods (over 5 seeds) towards different intervention samples. ($T = 33$ rounds, $|D_{int}^I| = 32$, $N = 1056$)

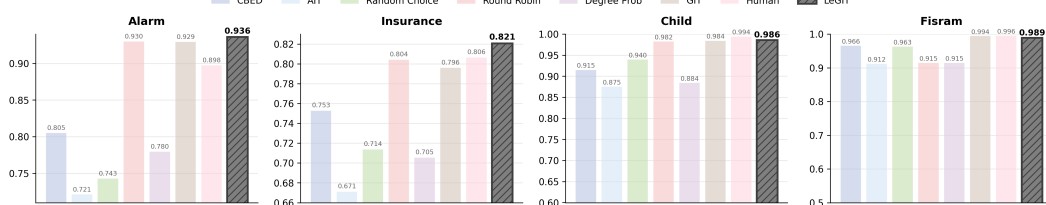

Figure 5: BSF metric for different methods (over 5 seeds) under Table 1 setting.

methods. In contrast, despite a faster decrease speed of GIT, GIT finally converges to a suboptimal solution due to unsuitable initialization, which verifies our discussion. Besides, SID results highlight LeGIT's robustness in preserving causal relationships and ensuring accurate causal inferences, which is essential in real-world applications. For BSF metrics in Fig. 5, higher values are indicative that the learned graph is more accurate and closely matches the true graph in terms of structure.

**Statistical significance.** In addition, we rerun the experiment under the Table 1 settings with 10 seeds on Alarm and Child dataset, and perform Paired T-test and Wilcoxon signed-rank test on SHD metrics (Virtanen et al., 2020) against LeGIT and GIT. The results are shown in Fig. 6. LeGIT exhibits the same level of performance as with 5 seeds, consistently outperforming GIT across all three metrics. This performance gap is statistically significant, with all p-values falling well below the 0.05 threshold for both datasets, confirming that our model's advantage is not an artifact of random chance. The full results are provided in Appendix C.4.

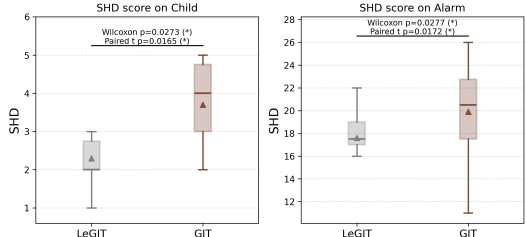

Figure 6: SHD score on Child and Alarm Dataset for LeGIT and GIT with P-values over 10 seeds.

The consistently low SHD and SID scores, with high BSF values, underscore the efficacy of LeGIT in accurately learning network structures and providing tangible benefits. Compared to heuristic-based methods like Random Choice and Round Robin, LeGIT offers a more strategic and data-driven approach, leading to better performance metrics.

Table 2: Average SHD and SID with standard deviation (over 5 seeds) for real-world data with a low data budget ($T = 33$ rounds, $|D_{int}^I| = 16$, $N = 528$).

| Methods | Alarm (37 Nodes, 46 Edges) | | Insurance (27 Nodes, 52 Edges) | | Child (20 Nodes, 25 Edges) | | Fisram (20 Nodes, 23 Edges) | |
|---|---|---|---|---|---|---|---|---|
| | SHD↓ | SID↓ | SHD↓ | SID↓ | SHD↓ | SID↓ | SHD↓ | SID↓ |
| CBED | 32.40 ±4.36 | 214.20 ±69.73 | 26.40 ±3.56 | 327.00 ±38.46 | 9.20 ±3.25 | 46.60 ±18.49 | 5.60 ±1.02 | 44.20 ±21.25 |
| AIT | 41.20 ±5.49 | 270.00 ±29.61 | 37.00 ±12.26 | 421.40 ±82.68 | 10.00 ±3.29 | 73.40 ±45.64 | 12.60 ±1.96 | 80.60 ±13.81 |
| Random Choice | 40.80 ±2.71 | 236.40 ±12.31 | 25.60 ±2.24 | 311.00 ±22.17 | 8.20 ±2.32 | 51.60 ±33.15 | 5.80 ±2.79 | 46.00 ±22.92 |
| Round Robin | 33.60 ±7.34 | 169.00 ±35.69 | 22.60 ±3.72 | 269.20 ±44.37 | 4.60 ±2.42 | 32.40 ±24.25 | 5.00 ±1.55 | 45.60 ±22.60 |
| Degree Prob | 42.60 ±6.34 | 244.20 ±35.06 | 31.80 ±4.40 | 351.00 ±27.64 | 9.00 ±2.90 | 60.80 ±23.01 | 11.00 ±3.74 | 67.80 ±20.45 |
| GIT | 27.20 ±4.71 | 177.80 ±61.65 | 22.40 ±3.72 | 296.00 ±44.23 | 6.00 ±1.55 | 33.80 ±15.75 | 3.60 ±3.61 | 35.20 ±31.90 |
| **Human** | 24.00 ±2.28 | 188.40 ±27.09 | 20.40 ±2.65 | 280.60 ±26.04 | 4.60 ±2.06 | 20.60 ±24.81 | 3.80 ±1.47 | 33.80 ±15.35 |
| **LeGIT** | 21.00 ±2.37 | 159.40 ±26.81 | 18.20 ±1.17 | 259.00 ±66.69 | 4.40 ±2.15 | 28.20 ±15.03 | 2.20 ±1.17 | 29.00 ±17.30 |

## 5.3 Low Data Experiment Analysis

Furthermore, we conduct additional experiments in an extremely low-data setting, where only 16 interventional data samples are sampled from each round, and other settings are the same as above.

This low-data setting is more practically relevant. Additionally, due to the insufficient intervention data, the performance of causal discovery algorithms in estimating effects is diminished (Lippe et al., 2022), which further tests the effectiveness and robustness of the intervention strategy.

The results presented in Table 2, where LeGIT achieves larger improvements under the challenging low-data condition across all datasets. We also provide the SHD curves and BSF metrics with respect to different intervention samples under Table 2 settings in Appendix C.5. These findings serve as strong evidence that reaffirm the effectiveness of LeGIT in real-world experimental design scenarios, where both the number of interventions and the sample size are limited.

The result of the low-data experiment further verifies our discussion that numerical methods suffer from noise or insufficient data, leading to a suboptimal solution. The numerical-based method does not even outperform round-robin on 3 smaller datasets, underscoring its limitations in such scenarios. In contrast, the use of LLMs enables scalable and effective guidance that complements numerical methods, reducing the risk of suboptimal convergence and having more stable performance in real-world applications.

### 5.4 DETAILED COMPARISONS AND ANALYSES

**Paired Up with Other Method.** To further test our design, we pair LeGIT with CBED and evaluate under the Table 1 settings. Table 3 shows that LeGIT with CBED reduces both SHD and SID on *Alarm* and *Child* when compared to vanilla CBED, indicating more accurate structure recovery and stronger interventional consistency at the same budget. These results confirm that LeGIT can be plugged into a numerical method and deliver consistency gains.

Table 3: Results of paired with CBED methods under Table 1 settings with 5 seeds.

| Dataset | Metric | Method | |
|---------|--------|--------|--------|
| | | LeGIT (CBED) | CBED |
| Alarm | SHD ↓ | **26.20** ± 4.17 | 28.20 ± 4.31 |
| | SID ↓ | **144.40** ± 24.21 | 213.80 ± 42.44 |
| | BSF ↑ | 0.7908 ± 0.07 | **0.8053** ± 0.06 |
| Child | SHD ↓ | **2.40** ± 1.02 | 5.40 ± 2.06 |
| | SID ↓ | **41.00** ± 9.85 | 44.40 ± 18.51 |
| | BSF ↑ | **0.9778** ± 0.02 | 0.9150 ± 0.05 |

**Compared to humans.** While Human interventions remain strong competitors, LeGIT bridges the gap between automated methods and expert-driven processes. LeGIT demonstrates superior performance on two complex datasets: Alarm and Insurance. As the number of variables increases, determining the optimal interventions to reveal the structure of the causal graph becomes combinatorially explosive. For humans, this process can be *extremely tedious or error-prone*, as they may subjectively favor certain nodes, failing to synthesize different viewpoints due to simpler mental models. In contrast, refer to Fig. 1, LLMs follow the instructions provided in Fig. 3 step by step and align them with their background knowledge. With the self-consistency prompt technique, LLMs generate more robust results, providing a highly cost-effective alternative to hiring multiple human experts for advice.

**Discussion.** LLMs' primary value lies in scalability and availability, providing immediate, cost-effective guidance in real-time, especially for online causal discovery where rapid interventions are required. They excel in large-scale systems with many variables, where it's infeasible for experts to assess all nodes. LLMs complement human oversight by filling gaps in availability, consistency, and knowledge while helping avoid expert biases. Additionally, LLMs quickly process metadata, saving experts time and providing a solid starting point, as seen in other AI-assisted tasks.

## 6 CONCLUSIONS

In this work, we investigated the feasibility of incorporating LLMs into the intervention targeting experimental design in causal discovery. We introduced a novel framework called LeGIT, which combines the best of previous numerical-based approaches and the rich knowledge in LLMs. Specifically, LeGIT leverages LLMs to warm up the online causal discovery procedure by identifying the influential root cause variables to begin the intervention. After setting up a rough skeleton of the underlying causal graph, LeGIT then integrates the numerical-based methods to continue to select the intervention targets. Empirically, we verified the effectiveness of LeGIT leveraging LLMs to warm up the online causal discovery can achieve the state-of-the-art performance across multiple realistic causal discovery benchmarks. Notably, LeGIT also outperforms humans in intervention targeting, highlighting the high potential and strong effectiveness of LeGIT. The findings with LeGIT demonstrate that LLMs offer a scalable and cost-efficient approach to enhance experimental design, paving the way for new research directions in causal analysis and scientific discovery.

## ETHICS STATEMENT

This work follows the ICLR Code of Ethics. This work mainly focuses on leveraging LLMs to better select the intervention targets for broader applications and social benefits. Besides, this paper does not raise any ethical concerns. This study does not involve any human subjects, practices, or data set releases, potentially harmful insights, methodologies, and applications, potential conflicts of interest and sponsorship, discrimination/bias/fairness concerns, privacy and security issues, legal compliance, and research integrity issues.

## REPRODUCIBILITY STATEMENT

All datasets used in our work are publicly available in the Bnlearn Repository (Scutari, 2010; BNMA). Our methods are fairly straightforward, and implementation details are already included in our paper descriptions in Sec. 3, Appendix C, and Appendix E.

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

## A  THE USE OF LARGE LANGUAGE MODELS

We use LLM to assist and polish our writing. At the same time, this paper primarily investigates whether we can leverage LLMs to identify intervention targets for obtaining interventional data for causal discovery.

## B  MORE DETAILS OF DATASETS

In this part, we will further introduce the 4 different domain Causal graph discovery datasets from bnlearn Repository (Scutari, 2010; Elidan, 2025) and BMNA BN Repository (BNMA). The adopted causal graphs are already among the largest compared to other works using LLMs for causal discovery (Zečević et al., 2023; Jin et al., 2023; Zhang et al., 2023a), as well as those used in numerical methods for online causal discovery (Olko et al., 2023). For the description of each variable, we refer to the original papers of each dataset, the bnlearn Package Document (Scutari, 2010). We show the ground truth and the out-degree node distributions as follows. For variable description, please refer to Appendix E.

**Fisram** (Freshwater Fish Injurious Species Risk Assessment Model), shown as Fig. 7, is to assess the potential invasiveness and harm of introduced freshwater fish species, aiding decisions on their importation. The model consists of 20 nodes and 23 edges, representing key species traits, environmental factors, and historical data used to assess potential ecological harm (Marcot et al., 2019).

**Child** show as Fig. 8 is used to model the diagnosis of pediatric health issues, particularly those that can occur in newborns or young children. It's often employed in studies related to decision support systems, where probabilistic graphical models assist in medical diagnosis, with 20 nodes and 25 edges (Dempster, 1993).

**Insurance** shown as Fig. 9 is intended to simulate a situation in which an insurance company needs to assess various risks and make decisions regarding policies, claims, and customer behavior. It represents the interdependencies between multiple insurance factors. It has 27 nodes and 52 edges (Binder et al., 1997).

**Alarm** shown as Fig. 10 is known as the ALARM (A Logical Alarm Reduction Mechanism) network, and it was originally developed to model a patient monitoring system for anesthesia purposes. It helps in predicting physiological conditions of patients, detecting potential complications, and generating alerts when necessary, consists of 37 nodes and 46 edges (Beinlich et al., 1989).

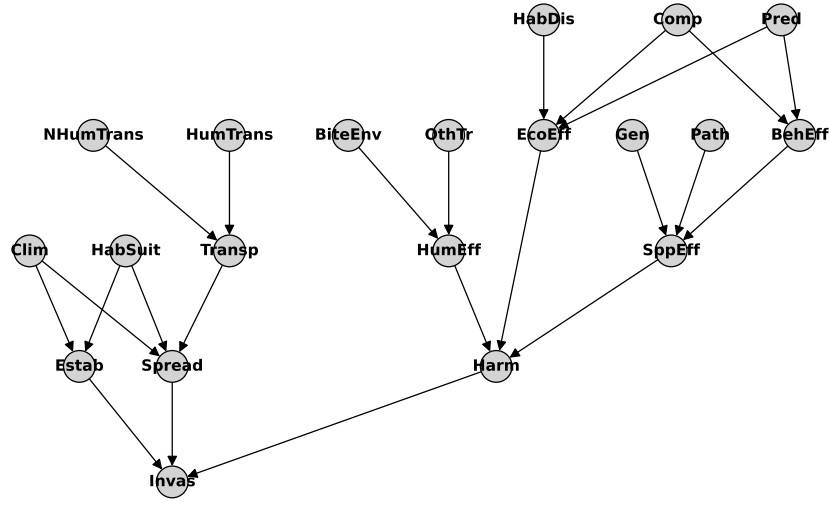

Figure 7: Ground truth Causal Graph for Fisram data.

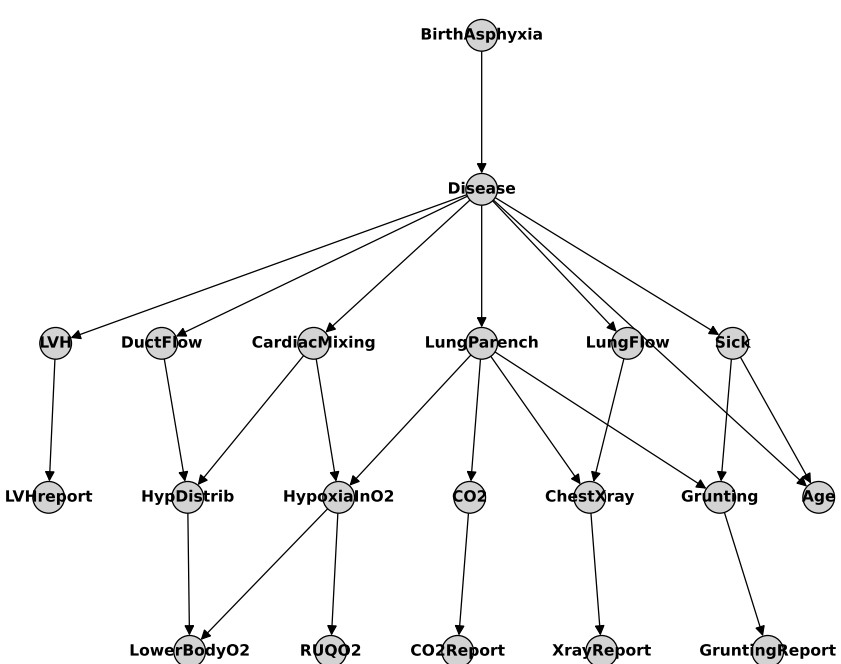

Figure 8: Ground truth Causal Graph for Child data.

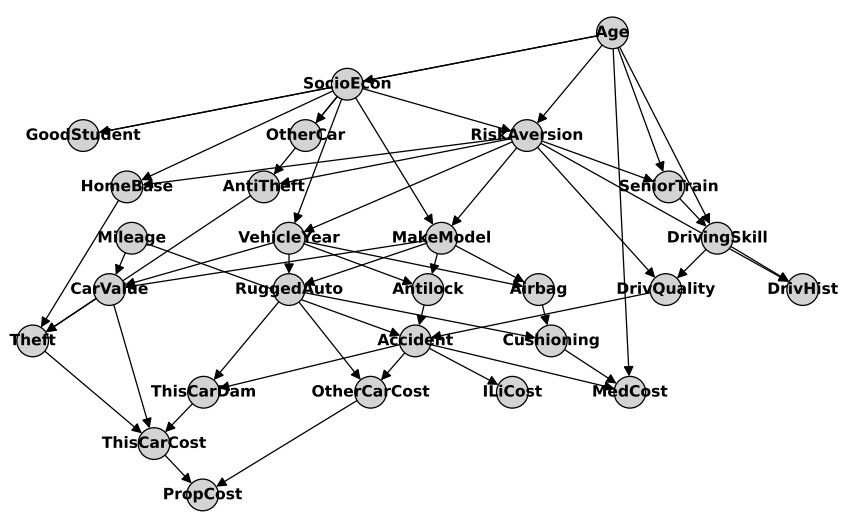

Figure 9: Ground truth Causal Graph for Insurance data.

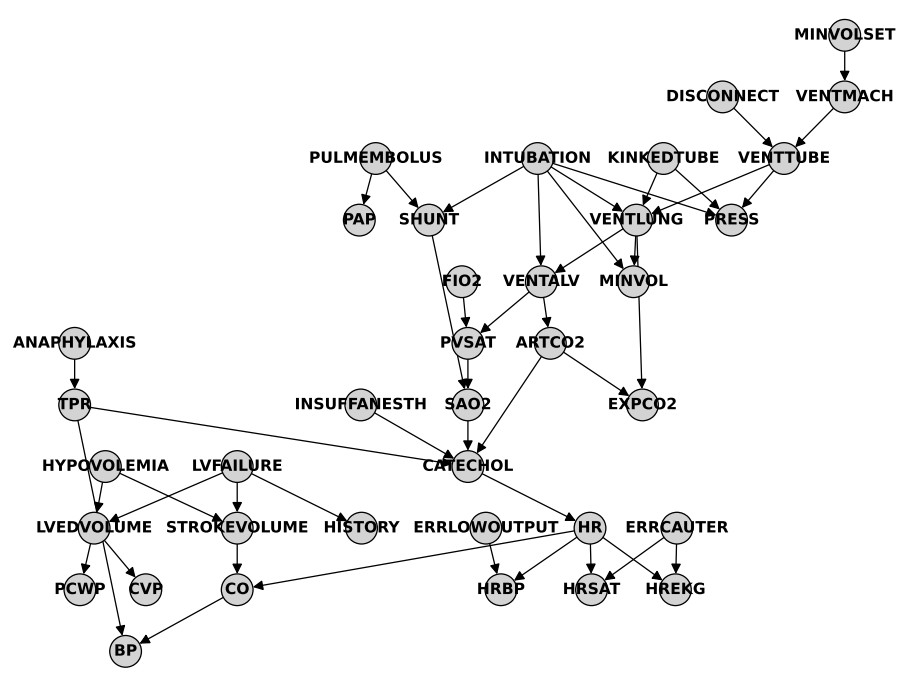

Figure 10: Ground truth Causal Graph for Alarm data.

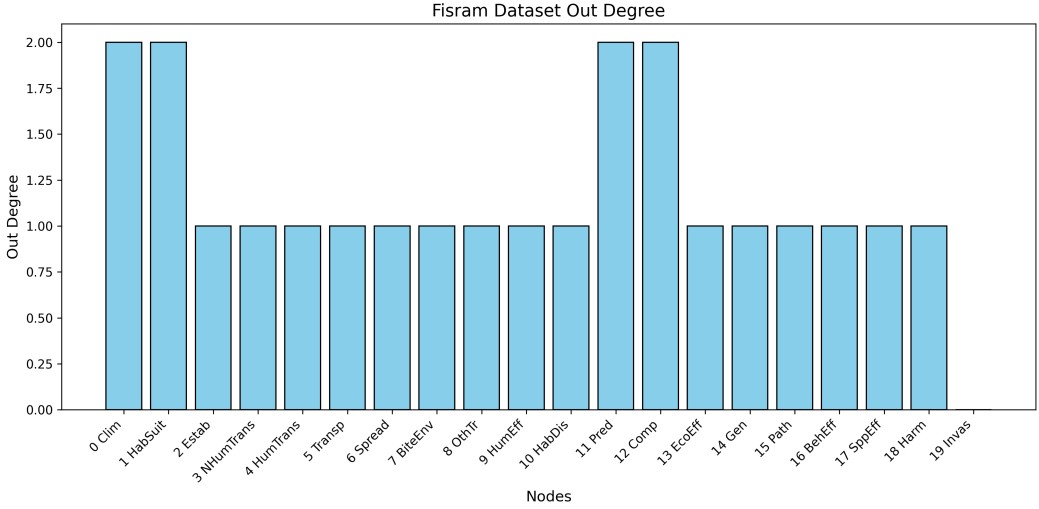

Figure 11: Out-degree distribution of Fisram data.

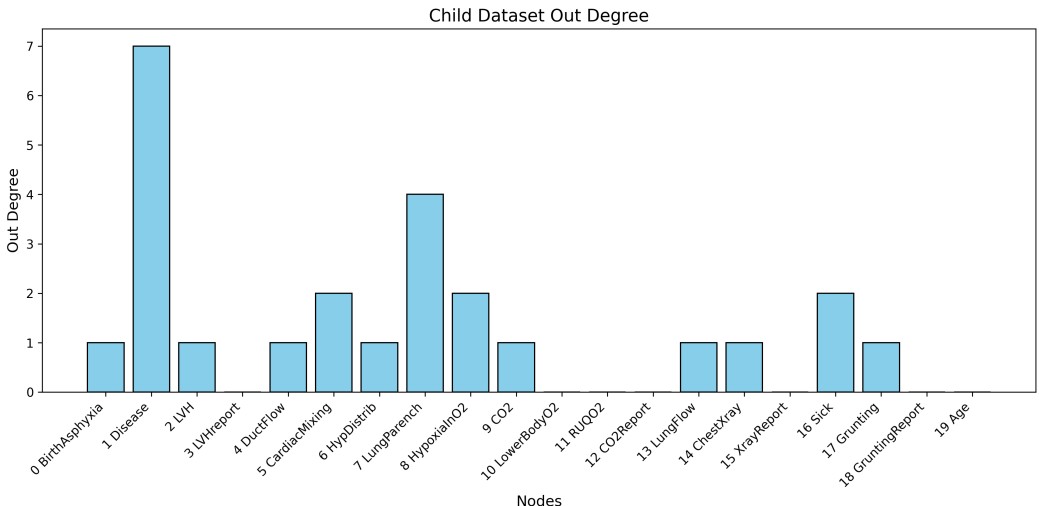

Figure 12: Out-degree distribution of Child data.

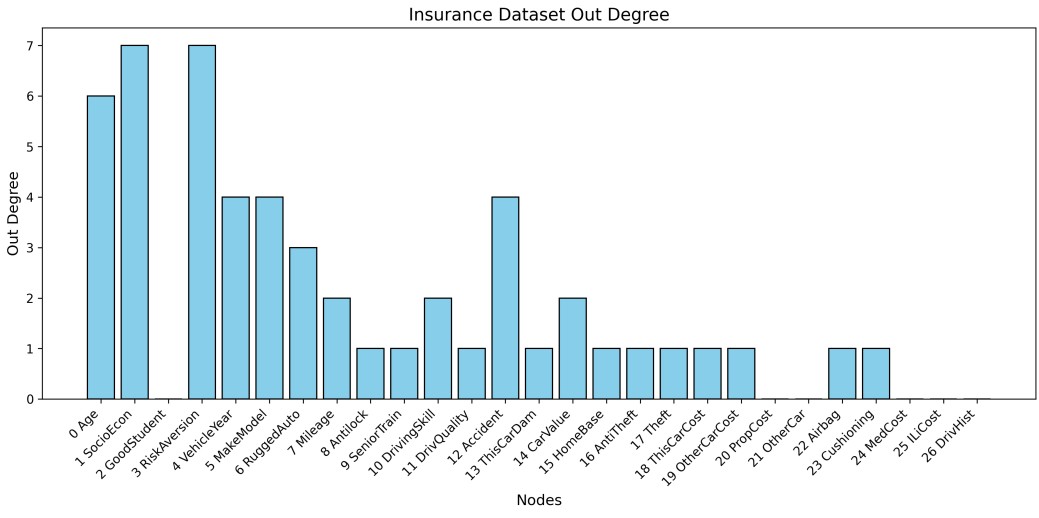

Figure 13: Out-degree distribution of Insurance data.

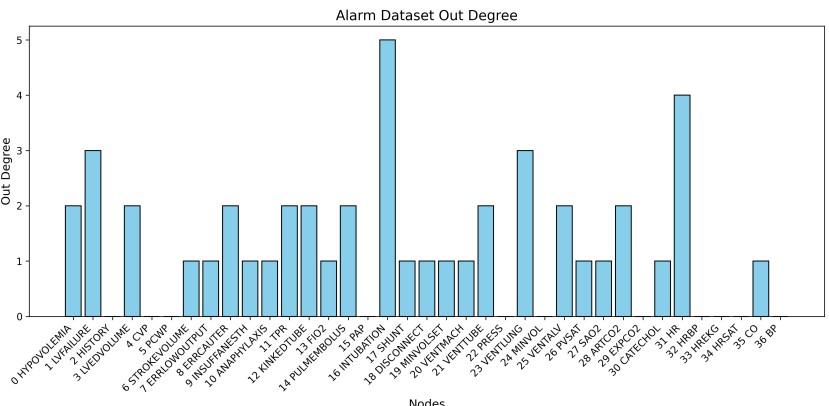

Figure 14: Out-degree distribution of Alarm data.

## C MORE DETAILS OF EXPERIMENTS

### C.1 ENCO HYPERPARAMETERS

For experiments using the ENCO framework, we used the exact parameters reported by (Lippe et al., 2022). These parameters are provided in Table 4 to ensure the completeness of our report.

Table 4: Hyperparameters used for the ENCO framework.

| PARAMETER | VALUE |
|---|---|
| SPARSITY REGULARIZER $\lambda_{sparse}$ | $4 \times 10^{-3}$ |
| DISTRIBUTION MODEL | 2 LAYERS, HIDDEN SIZE 64, LEAKYRELU($\alpha = 0.1$) |
| BATCH SIZE | 128 |
| LEARNING RATE - MODEL | $5 \times 10^{-3}$ |
| WEIGHT DECAY - MODEL | $1 \times 10^{-4}$ |
| DISTRIBUTION FITTING ITERATIONS F | 1000 |
| GRAPH FITTING ITERATIONS G | 100 |
| GRAPH SAMPLES K | 100 |
| EPOCHS | 30 |
| LEARNING RATE - $\gamma$ | $2 \times 10^{-2}$ |
| LEARNING RATE - $\theta$ | $1 \times 10^{-1}$ |

### C.2 DETAILED METRICS

In this section, we present the details of 3 different metrics mentioned in the experiment part.

- The Structural Hamming Distance (SHD) (Tsamardinos et al., 2006): SHD is a frequently employed score comparing graph structures via their binary adjacency matrices. It represents the minimum sum of edge additions ($A$), deletions ($D$), and reversals ($R$) required to convert one adjacency matrix into that of the ground truth causal graph.

$$SHD = A + D + R \qquad (1)$$

- Structural Intervention Distance (SID) (Peters & Bühlmann, 2015): This metric measures how closely two DAGs, $\mathcal{G}$ and $\mathcal{H}$, align in terms of the causal effects they encode. SID is defined as the total count of intervention distributions (from node i to node j) that are inaccurately predicted by the candidate graph H when compared against the reference graph G. Consequently, SID reveals the impact of edge errors within H on the resulting causal effect estimations.

$$SID = \#\{(i,j), i \neq j \mid \text{the intervention distribution from}$$
$$i \text{ to } j \text{ is falsely estimated by } \mathcal{H} \text{ with respect to } \mathcal{G}\}. \qquad (2)$$

- Balanced Scoring Function (BSF) (Constantinou, 2019): BSF offers an unbiased method for evaluating the performance of graph structure learning algorithms. It achieves this by normalizing the contributions of true positives ($TP$), true negatives ($TN$), false positives ($FP$), and false negatives ($FN$) according to the prevalence of actual dependencies and independencies in the reference graph structure. The calculation is performed as follows:

$$\text{BSF} = \frac{1}{2}\left(\frac{\text{TP}}{a} + \frac{\text{TN}}{i} - \frac{\text{FP}}{i} - \frac{\text{FN}}{a}\right), \tag{3}$$

Here, $a$ is the count of arcs in the ground truth graph. The term $i$ corresponds to the number of absent arcs (independencies) in the true graph, calculated as $i = \frac{|N|\times(|N|-1)}{2} - a$, where $|N|$ is the total count of nodes.

## C.3 Open-source models experiments

We tested DeepSeek-R1-Distill-Qwen-14B (DeepSeek-AI, 2025), Deepseek-V3 (DeepSeek-AI, 2024), and LLama-3.1-405B (AI, 2024) in the Alarm and Insurance datasets. The SHD and MeanNHD (NHD = $\frac{1}{N_{node}^2}SHD$) results for the Alarm and Insurance datasets within 5 random seeds in 2 settings are shown in Table 5. With different LLMs, we can find that LeGIT still consistently shows strong performance with these LLMs, highlighting the robustness and adaptability of our proposed framework.

Table 5: SHD and MeanNHD of using different LLMs in LeGIT framework under 2 settings with 5 random seeds.

| Methods | Normal Settings | | | Low Data Settings | | |
|---|---|---|---|---|---|---|
| | Alarm | Insurance | MeanNHD ↓ | Alarm | Insurance | MeanNHD ↓ |
| GIT | 19.60 ± 3.77 | 16.40 ± 3.14 | 0.0184 | 27.20 ± 4.71 | 22.40 ± 3.72 | 0.0253 |
| **LeGIT (GPT-4O)** | **17.40 ± 3.61** | **12.60 ± 0.80** | **0.0161** | **21.00 ± 2.37** | **18.20 ± 1.17** | **0.0202** |
| LeGIT (DeepseekR1-14B) | 20.00 ± 2.12 | 14.20 ± 2.71 | 0.0170 | 22.00 ± 0.82 | 24.60 ± 2.15 | 0.0249 |
| LeGIT (DeepseekV3) | 18.60 ± 3.44 | 14.60 ± 2.72 | 0.0168 | 22.40 ± 1.85 | 20.60 ± 1.02 | 0.0223 |
| LeGIT (LLama-3.1-405B) | 18.60 ± 2.33 | 15.80 ± 4.07 | 0.0176 | 28.00 ± 4.15 | 18.20 ± 4.26 | 0.0227 |

## C.4 Statistical Significance

The full results on Alarm and Child datasets under Fig. 7 are provided in Table 6.

Table 6: Results on Alarm and Child datasets under Table 1 settings with 10 seeds, and P-value of the Paired T-test and the Wilcoxon test with GIT and LeGIT SHD.

| Dataset | Metric | Method | | Test on SHD | |
|---|---|---|---|---|---|
| | | LeGIT | GIT | Paired T | Wilcoxon |
| Alarm | SHD ↓ | 17.60 ± 2.76 | 19.90 ± 4.37 | | |
| | SID ↓ | 122.00 ± 35.40 | 140.60 ± 52.36 | 0.0172 | 0.0277 |
| | BSF ↑ | 0.9293 ± 0.03 | 0.9272 ± 0.02 | | |
| Child | SHD ↓ | 2.30 ± 1.00 | 3.70 ± 1.10 | | |
| | SID ↓ | 23.70 ± 11.99 | 30.20 ± 18.02 | 0.0165 | 0.0273 |
| | BSF ↑ | 0.9655 ± 0.03 | 0.9584 ± 0.03 | | |

We also perform the Paired T-test and the Wilcoxon Signed-Rank test (Virtanen et al., 2020) against GIT and LeGIT results in Table 1. The result is shown in Table 7. Considering the sample size, at a 90% confidence level, we believe that the results of LeGIT outperform GIT on all four datasets.

The results clearly indicate that LeGIT outperforms existing baseline methods across all three evaluation metrics.

## C.5 Low-Data settings Visualization

In this section, we provide the SHD curve under low-data settings as Fig. 15 and the selected Node Frequency obtained by different strategies on Epoch 0-4 as Fig. 17. And the BSF metrics in Fig. 16.

Table 7: P-value of Paired T-test and Wilcoxon test with GIT and LeGIT SHD results in Table 1.

|  | Paired T-test | Wilcoxon test |
|---|---|---|
| Alarm | 0.074 | 0.100 |
| Insurance | 0.060 | 0.067 |
| Child | 0.070 | 0.083 |
| Fisram | 0.099 | 0.102 |

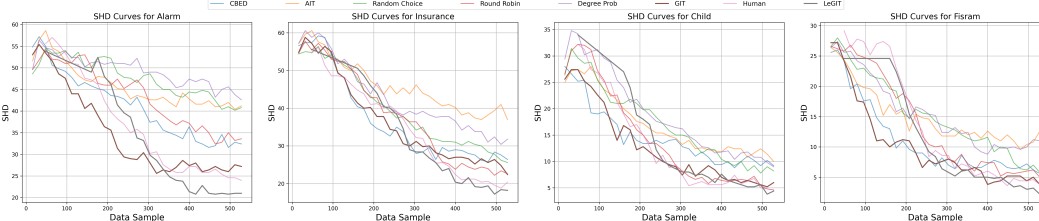

Figure 15: SHD metric for different methods (over 5 seeds) under Table 2 setting ($T = 33$ rounds, $|D_{int}^I| = 16, N = 528$)

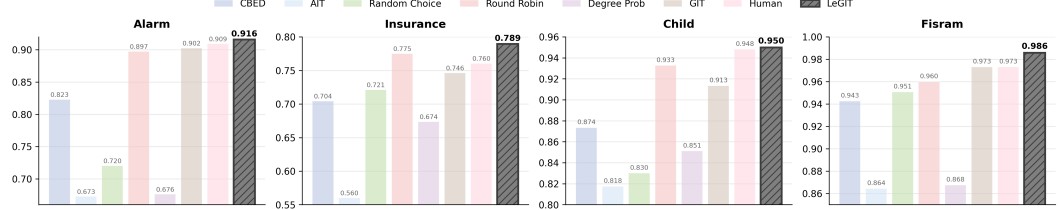

Figure 16: BSF metric for different methods (over 5 seeds) towards different intervention samples. ($T = 33$ rounds, $|D_{int}^I| = 32, N = 1056$)

## C.6 RESOURCES

We utilized a system comprising two Intel Xeon Platinum 8358P processors with 2.6GHz, two NVIDIA A40 GPUs (48GB each), and 1 TB of memory. For the large language model (LLM) API, we leveraged the Azure platform.

## C.7 FINAL CAUSAL GRAPH

In this section, we present the final causal graph after $T = 33$, total sample $N = 1056$ results with GIT, Human, and LeGIT.

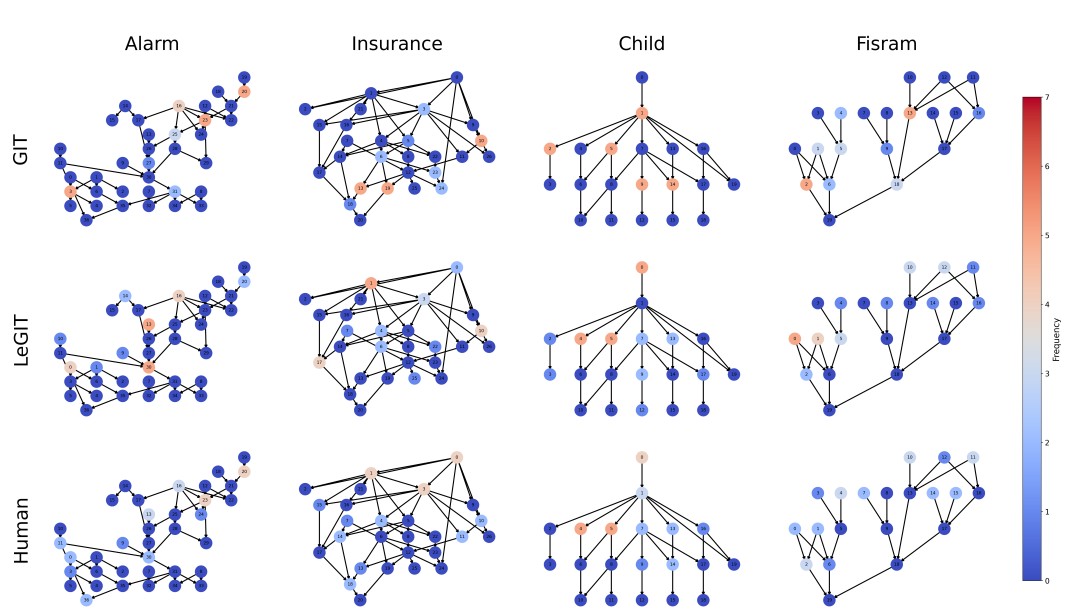

Figure 17: The selected Node Frequency obtained by different strategies on Epoch 0-4 under Table 2 setting.

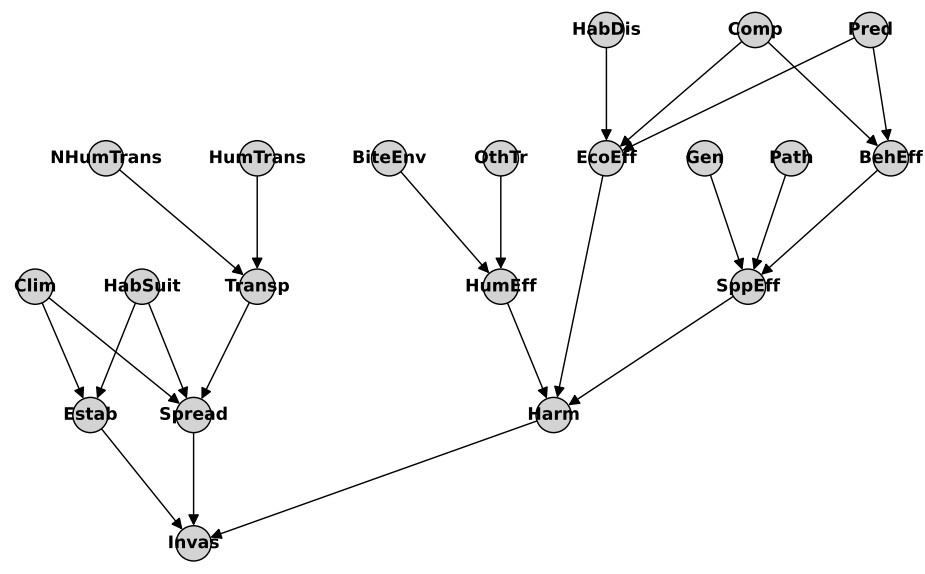

Figure 18: LeGIT final causal graph for Fisram dataset

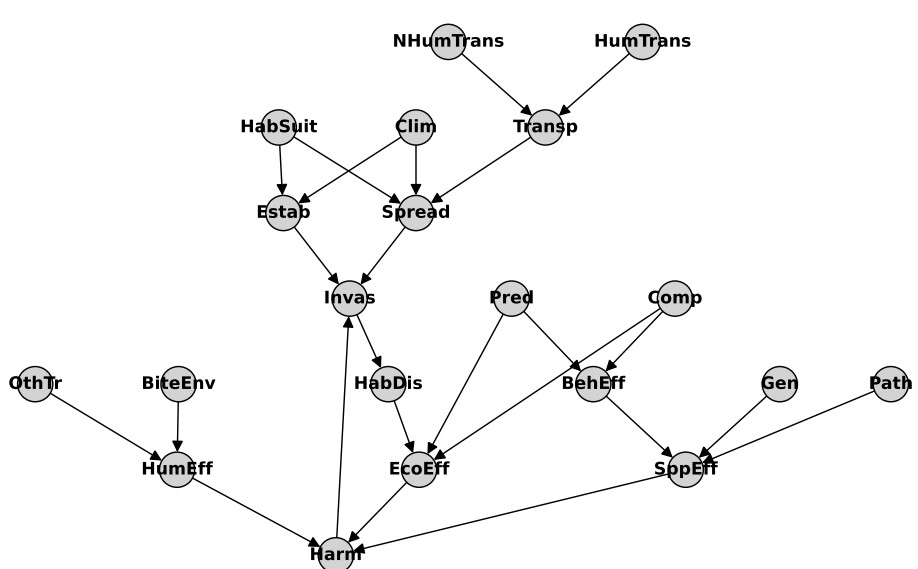

Figure 19: GIT's final causal graph for Fisram dataset

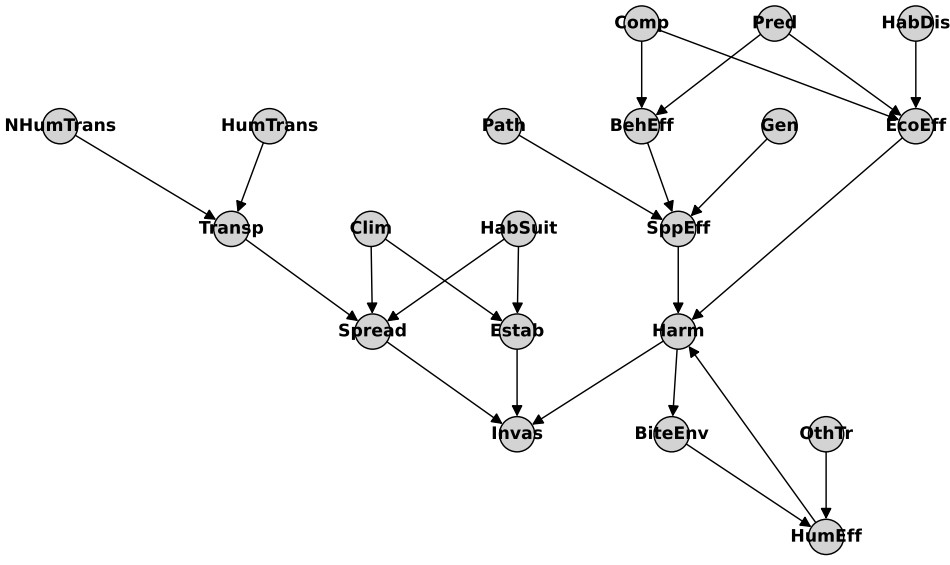

Figure 20: Human's final causal graph for Fisram dataset

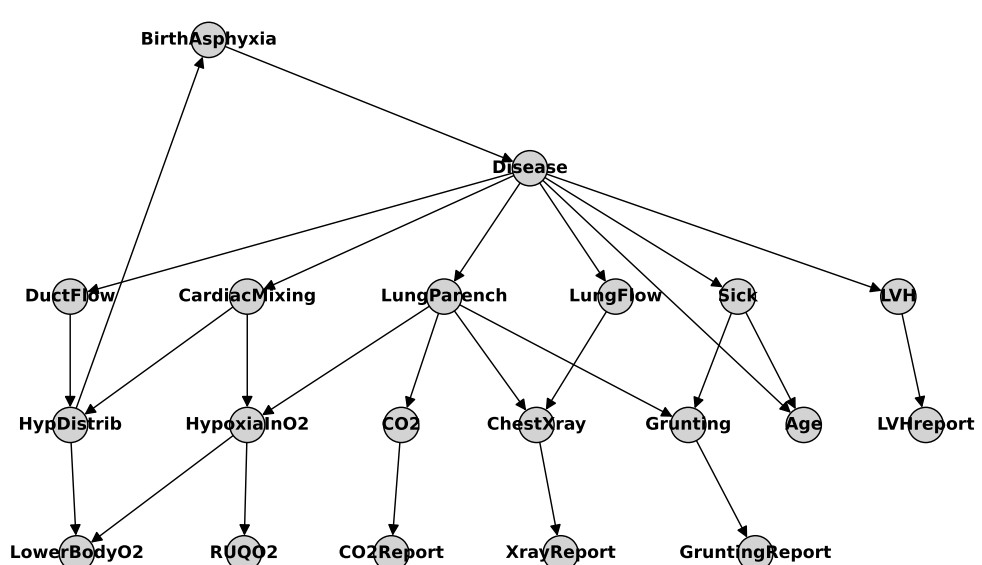

Figure 21: LeGIT final causal graph for Child dataset

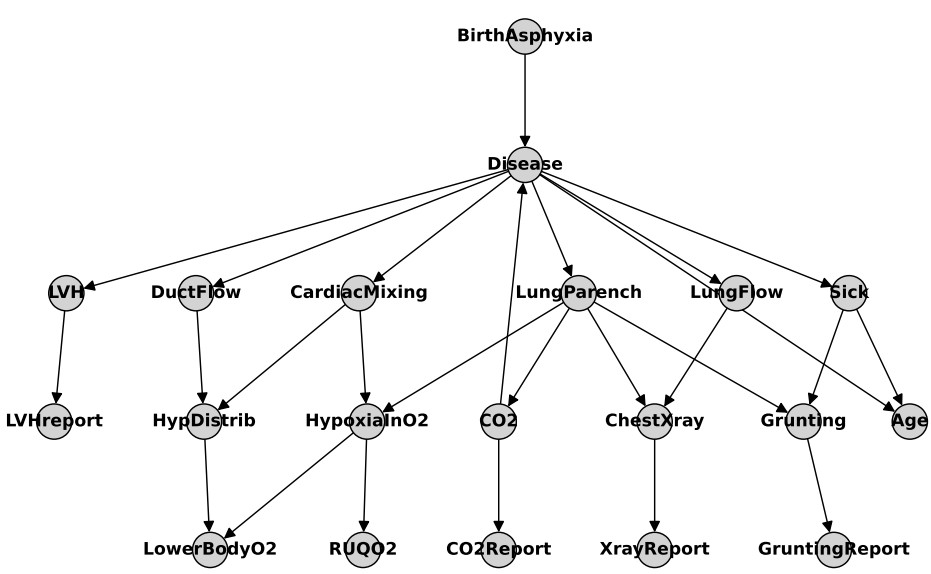

Figure 22: Human's final causal graph for child dataset

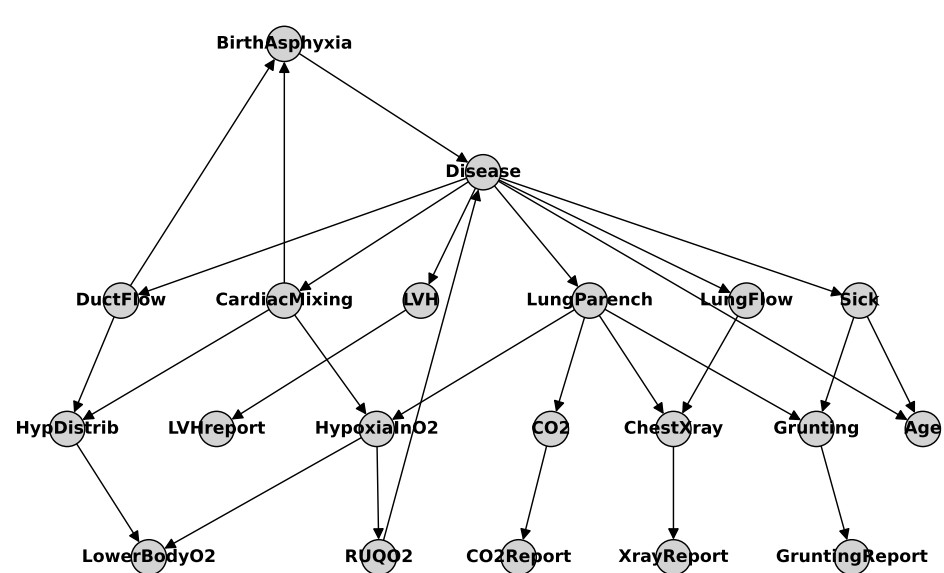

Figure 23: GIT's final causal graph for Child dataset

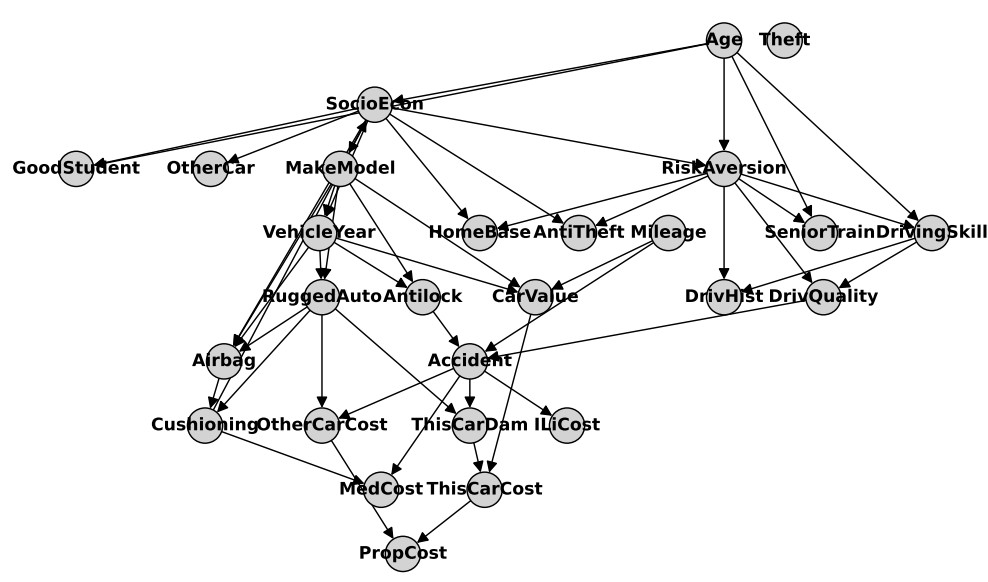

Figure 24: LeGIT final causal graph for Insurance dataset

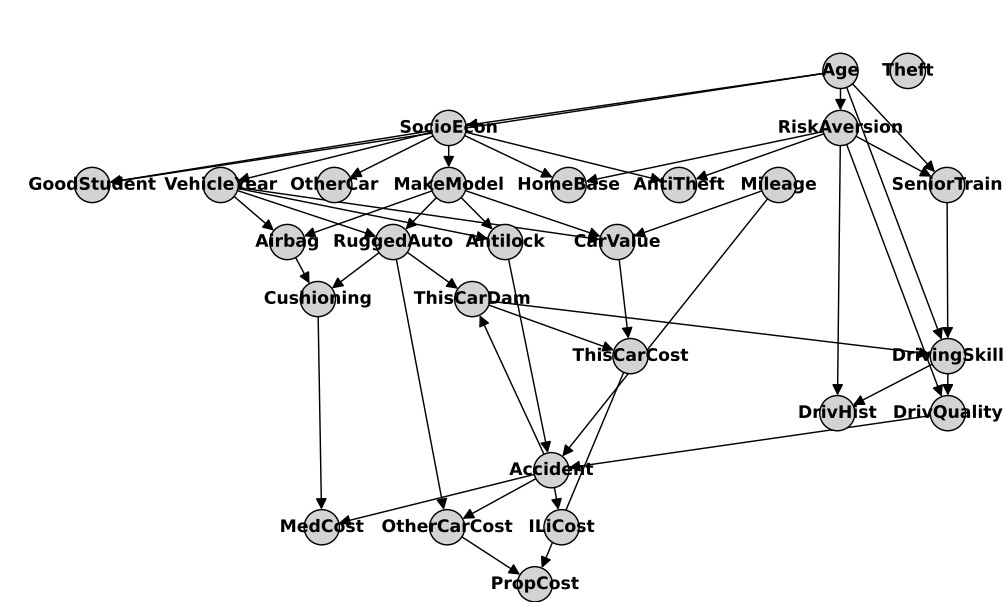

Figure 25: Human's final causal graph for Insurance dataset

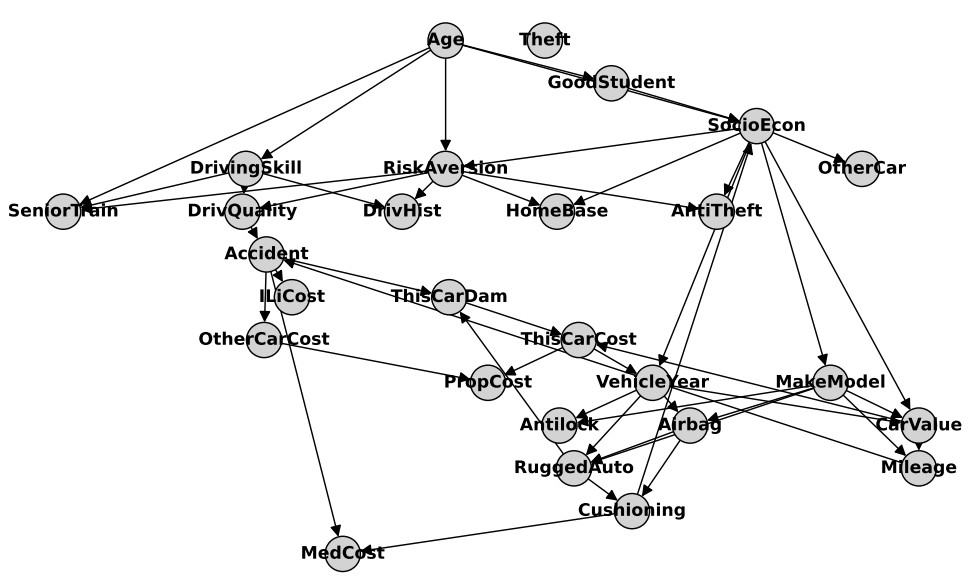

Figure 26: GIT's final causal graph for Insurance dataset

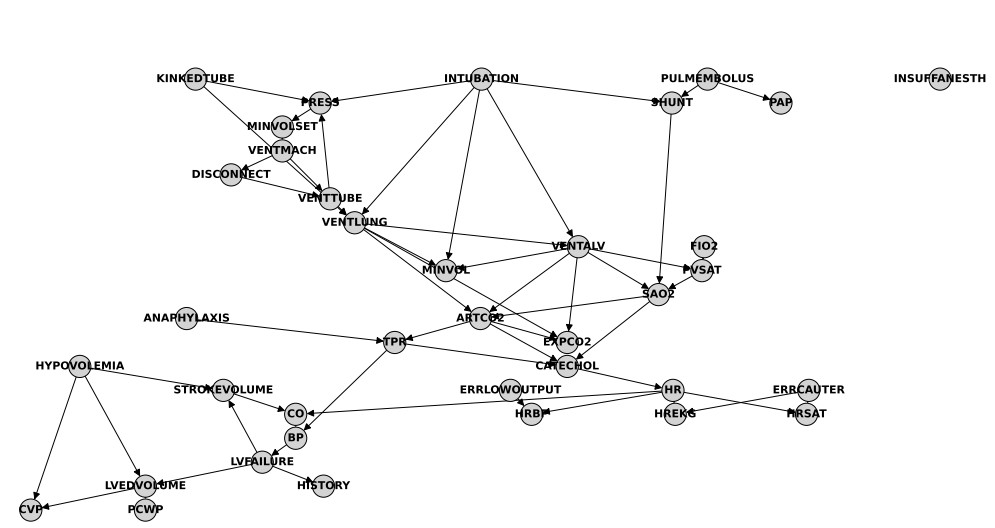

Figure 27: LeGIT final causal graph for Alarm dataset

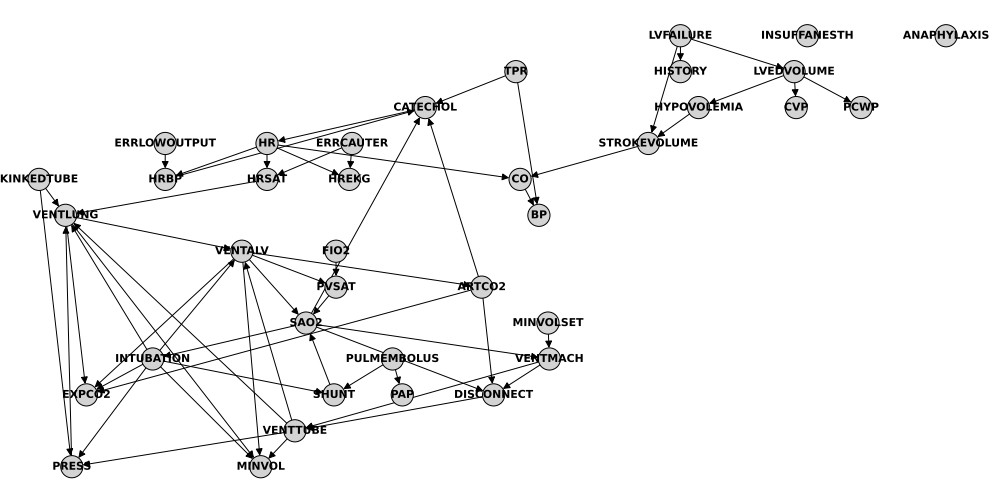

Figure 28: Human's final causal graph for Alarm dataset

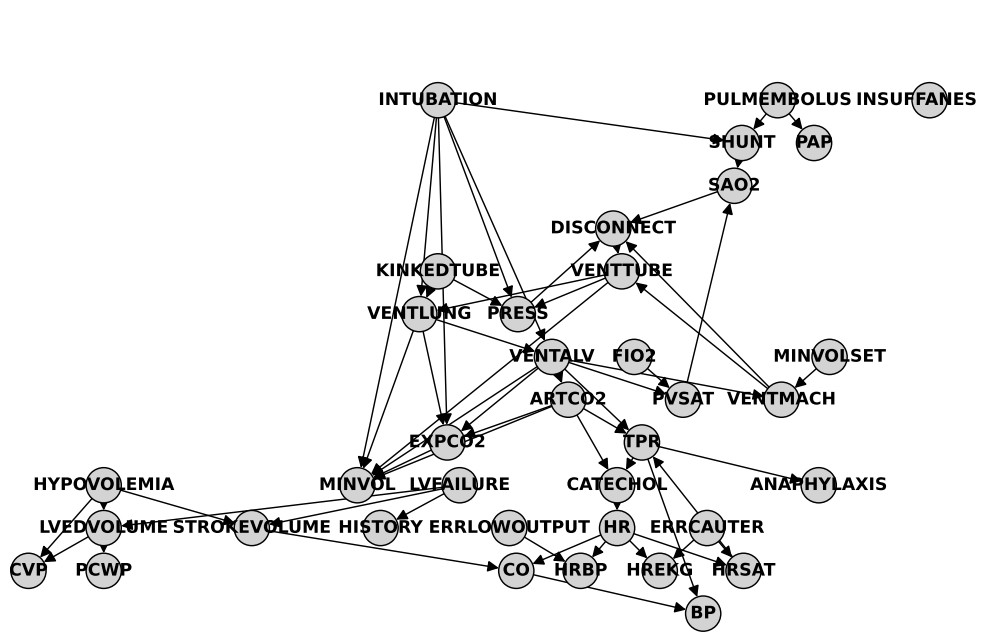

Figure 29: GIT's final causal graph for Alarm dataset

## D  CONVERGENCE OF CAUSAL DISCOVERY WITH LEGIT

In this section, we provide a convergence argument for LeGIT, which combines a Large Language Model (LLM) warmup phase with a numerical-based intervention targeting strategy (e.g., GIT (Olko et al., 2023)).

### D.1  PRELIMINARIES AND NOTATION

**Structural Causal Models and Online Causal Discovery.**    We use standard definitions of structural causal models (SCMs), directed acyclic graphs (DAGs), and single-node (hard) interventions as in, e.g., ENCO (Lippe et al., 2022). Let

$$G^* = (V, E^*), \qquad V = \{1, 2, \ldots, n\},$$

indexing causal variables $(X_1, \ldots, X_n)$; the goal is to recover $G^*$. In an online setting, at each round $t = 1, 2, \ldots, T$, we choose a single intervention target $I_t \in V$ and obtain a small batch of interventional samples under the regime $do(X_{I_t})$. We write $M_i$ for the interventional setting $do(X_i)$ and use $p(\cdot \mid M_i)$ for the corresponding interventional distribution. Newly acquired interventional data are then used to update the current structural hypothesis and its parameters.

**LEGIT and the Warmup Stage.**    LEGIT begins with a small number of *warmup* rounds $T_{\text{warmup}}$, optionally followed by a *bootstrap* stage $T_{\text{bootstrapped}}$. During these initial stages, an LLM proposes intervention targets based on domain descriptions or meta-information about variables. After warmup and bootstrap, we perform a single *re-sampling* pass that revisits both the initial LLM list and the bootstrapped list exactly once (to mitigate outliers and ensure coverage). The algorithm then switches to a purely numerical strategy for intervention selection—for example, GIT (Olko et al., 2023) or a Bayesian method (Brouillard et al., 2020). Formally,

$$I_t = \begin{cases} \text{LLM-based selection (warmup/bootstrapped),} & t \leq 2\big(T_{\text{warmup}} + T_{\text{bootstrapped}}\big), \\ \text{numerical-based selection,} & 2\big(T_{\text{warmup}} + T_{\text{bootstrapped}}\big) < t \leq T. \end{cases}$$

This schedule will be used in our analysis to justify the warmup coverage condition and subsequent convergence guarantees.

### D.2  CONVERGENCE PROOF

Throughout, let $G^\star$ denote the true DAG and let $\text{Pa}^\star(j)$ be the true parent set of node $j$.

**Assumptions.**    We adopt standard assumptions used by interventional CD methods such as GIT and ENCO:

- **A1 (Faithfulness).** The observational/interventional distributions are faithful to a unique DAG $G^\star$.

- **A2 (Sufficiency).** No latent confounding; all relevant variables are observed.

- **A3 (Convergent Base Method).** Given sufficiently many interventional samples on a suitable set of targets, the numerical backbone (e.g., GIT for target selection coupled with ENCO for parameter/structure updates) converges to $G^\star$; see original paper (Olko et al., 2023; Lippe et al., 2022) for formal statements.

In addition, we formalize what it means for an intervention to be *informative* for a candidate edge.

**Definition D.1** (Edge-informative single-node intervention)**.** *Fix an ordered pair $(i, j)$. Consider an intervention on $X_i$ and let $\widehat{\text{pa}}(j)$ denote the current candidate parent set for $j$. Define the interventional log-likelihood gap*

$$\Delta_{i \to j} := \mathbb{E}_{M_i}[\log p(X_j \mid \widehat{\text{pa}}(j) \cup \{X_i\}) - \log p(X_j \mid \widehat{\text{pa}}(j))],$$

*where $M_i$ indicates data drawn under $do(X_i)$. The intervention $do(X_i)$ is* informative *for $(i, j)$ if $\Delta_{i \to j} > 0$.*

This matches the quantity whose sign drives the expected gradient on the ENCO orientation parameter for $(i, j)$: when $i \in \mathrm{Pa}^\star(j)$, $\Delta_{i \to j}$ is strictly positive for at least one candidate $\widehat{\mathrm{pa}}(j)$ (ENCO's consistency condition), whereas if $i \notin \mathrm{Pa}^\star(j)$ the expectation is non-positive.[3]

**Warmup coverage.** Our LLM warmup plus re-sampling stage is designed to touch influential/parent candidates early. We capture this by a mild coverage requirement:

**Assumption D.2** (Warmup edge coverage). *There exists $\alpha > 0$ such that in each warmup epoch, with probability at least $\alpha$, the selected target $v$ is a true parent of some node $c$ whose $(v, c)$ orientation is not yet resolved. Equivalently, the warmup yields a non-empty set of edge-informative interventions in the sense of Def. D.1.*

This is satisfied when (i) the LLM list contains some true parents for currently unresolved children (as observed empirically), and (ii) the re-sampling stage visits all items in the LLM/bootstrapped lists at least once, preventing a single bad suggestion from dominating.

### D.2.1 KEY LEMMA: INFORMATIVE INTERVENTIONS MOVE ENCO TOWARD THE CORRECT ORIENTATION

**Lemma D.3** (Positive drift on true edges). *Fix $j$ and any true parent $i \in \mathrm{Pa}^\star(j)$. Under A1–A2, if $do(X_i)$ is taken and is informative for $(i, j)$ in the sense of Def. D.1, then the expected ENCO update on the orientation parameter for $(i, j)$ points toward $i \to j$. Consequently, after finitely many such informative interventions on distinct true parents across unresolved edges, the expected number of misoriented or spurious edges strictly decreases.*

*Proof.* When intervening on $X_i$, the natural dependence from $i$ to $j$ is disrupted in the data distribution. If $i \in \mathrm{Pa}^\star(j)$, incorporating $X_i$ as a parent of $j$ improves predictive adequacy for $X_j$ under $do(X_i)$ relative to omitting it, yielding $\Delta_{i \to j} > 0$. In ENCO, the expected gradient of the orientation parameter for $(i, j)$ under $M_i$ has the same sign as $\Delta_{i \to j}$; hence its expectation is positive and pushes the model toward the correct orientation $i \to j$. If $i \notin \mathrm{Pa}^\star(j)$, the gap is non-positive in expectation and the update does not reinforce a wrong arrow. Summing over a finite set of unresolved true edges receiving informative interventions produces a negative expected change in a potential such as the structural Hamming distance to $G^\star$. $\square$

### D.2.2 CONVERGENCE OF THE COMBINED PROCEDURE

**Theorem D.4** (Convergence of LEGIT). *Under A1–A3 and Assumption D.2, LEGIT converges to the true DAG $G^\star$ as the total number of acquisition rounds $T \to \infty$.*

*Proof.* By Assumption D.2 and Lemma D.3, the warmup/re-sampling stage executes a non-empty set of edge-informative single-node interventions, reducing the expected number of misoriented/extra edges around influential nodes. From round $t = 2(T_{\mathrm{warmup}} + T_{\mathrm{bootstrapped}}) + 1$ onward, target selection follows the base method (e.g., GIT), and parameters/structure are updated numerically (e.g., ENCO). By A3, once a sufficient variety of informative interventions has been gathered, the base method converges to $G^\star$.

Intuitively, warmup shrinks the hypothesis space by resolving high-impact ambiguities; the base method then continues to pick targets whose interventional data (together with the already-collected warmup data) satisfy the sufficient conditions required by its own convergence proof (Olko et al., 2023; Lippe et al., 2022). Therefore, the overall procedure converges to $G^\star$. $\square$

**Remark D.5** (Relation to single-node identifiability). *Our notion of informativeness is compatible with the classical single-node identifiability result: with faithfulness and no latent confounding, a finite set of single-node interventions suffices to identify the DAG up to exact orientation (Eberhardt et al., 2012). The warmup aims to hit true parents early (often high-degree or high-influence nodes), and the base method continues selecting targets; together, they assemble a set of interventions that meets the sufficient conditions for exact recovery used in A3.*

---

[3]ENCO updates edge-orientation parameters only when intervening on one of the endpoints; in expectation the update direction is proportional to the interventional likelihood contrast between "edge present" and "edge absent". See Lippe et al. (2022), Thm. 3.1 conditions.

**Remark D.6** (Extension to Other Methods). *Although we use GIT and ENCO as our illustrative example, any gradient-based or Bayesian active learning method for causal discovery that is guaranteed to converge given a suitable variety of interventions can replace GIT in LeGIT. Under the same conditions (A1–A3), the combined procedure will likewise converge to the true DAG $G^*$.*

# E  EXAMPLES OF PROMPTS

For robust performance, we actually shuffle the order of variable descriptions following the self-consistency prompt skill. We provide the prompt templates and the description of the variables used in LeGIT below.

---

**Fisram Warmup Prompt**

You are a helpful assistant and expert in Freshwater Fish Injurious Species Risk Assessment Model system research. Here are some tips that you can pay attention to:

1. Assess whether there is a direct causal relationship, and consider potential confounding variables that might affect the relationship that could potentially not causal relationship.

2. Distinguish between correlations and causation; verify that correlations are not mistaken for causal relationships.

3. Ensure the correct temporal order of variables; confirm that the cause precedes the effect. Assuming we can do interventions to all the variables, your job is to assist in designing the best intervention experiments among the following variables to help discover their causal relations:

<OthTr>: Non-bite/toxin traits posing human-health risks (e.g., zoonotic pathogens, physical injury from leaping species).

<Harm>: Actual or potential physical or behavioral injury to native species and/or humans, or damage to habitats. <BehEff>: Combined effect of predation and competition on native species behavior and viability.

<EcoEff>: Overall impact of habitat disturbance, predation, and competition on ecosystem structure and function.

<Clim>: Sum of counts for climate similarity scores 6–10 divided by the sum of all climate scores, as calculated by the CLIMATCH or RAMP tools.

<HumEff>: Combined influence of bites/toxins and other detrimental traits on humans.

<Estab>: Actual or potential for self-sustaining wild populations based on climate and habitat inputs.

<NHumTrans>: Dispersal assistance by natural agents (wind, water, animals) beyond the species' own movement.

<HumTrans>: Intentional or unintentional movement by humans (e.g., trade, ballast water, recreational stocking).

<BiteEnv>: Direct adverse effects on human health via bites, stings, toxins, injections, ingestion, or absorption. <Pred>: Capacity to prey on and negatively affect native species populations.

<Gen>: Capacity to affect native species' genetics via hybridization, GMO escape, or introgression.

<HabDis>: Capacity to modify or degrade habitat (erosion, eutrophication, sedimentation).

<Invas>: Final invasive-injurious outcome under the Lacey Act criteria, integrating Establishment, Spread, and Harm. <Comp>: Capacity to compete with native species for food, space, or habitat.

<SppEff>: Overall impact of predation, competition, and genetics on native species viability.

<Path>: Role in spreading infectious agents (bacteria, viruses, parasites, fungi) to native wildlife.

<HabSuit>: Degree to which available habitat in the potential introduction area matches the species' known habitats.

<Spread>: Actual or potential spatial expansion across ecosystems, driven by climate, habitat, and transport.

<Transp>: Combined human and non-human dispersal influence.

---

Assuming we can do interventions to all the variables, given the aforementioned variables and their descriptions, can you **echo your knowledge of those variables**, **temporally analyze** their relations, and then **choose the best 4 intervention targets from all the variables** which hopefully are the root causes of the other variables to start our analysis of their causal relations?
Let's think and analyze step by step. Then, provide your final answer (variable names only) within the tags <Answer>...</Answer>, separated by ", ".

### Child Warmup Prompt

You are a helpful assistant and expert in children's disease research. Here are some tips that you can pay attention to:
1. Assess whether there is a direct causal relationship, and consider potential confounding variables that might affect the relationship that could potentially not causal relationship.
2. Distinguish between correlations and causation; verify that correlations are not mistaken for causal relationships.
3. Ensure the correct temporal order of variables; confirm that the cause precedes the effect.
Assuming we can do interventions to all the variables, your job is to assist in designing the best intervention experiments among the following variables to help discover their causal relations:
<LungFlow>: low blood flow in the lungs
<ChestXray>: having a chest x-ray
<Disease>: infant methemoglobinemia
<Grunting>: grunting in infants
<Age>: age of infant at disease presentation
<XrayReport>: lung excessively filled with blood
<RUQO2>: level of oxygen in the right upper quadriceps muscle
<DuctFlow>: blood flow across the ductus arteriosus
<HypoxiaInO2>: hypoxia when breathing oxygen
<Sick>: presence of an illness
<CO2Report>: a document reporting high level of CO2 levels in blood
<LungParench>: the state of the blood vessels in the lungs
<LVH>: having left ventricular hypertrophy
<LowerBodyO2>: level of oxygen in the lower body
<BirthAsphyxia>: lack of oxygen to the blood during the infant's birth
<CO2>: level of CO2 in the body
<LVHreport>: report of having left ventri
<GruntingReport>: report of infant grunting
<CardiacMixing>: mixing of oxygenated and deoxygenated blood
<HypDistrib>: low oxygen areas equally distributed around the body
Assuming we can do interventions to all the variables, given the aforementioned variables and their descriptions, can you **echo your knowledge of those variables**, **temporally analyze** their relations, and then **choose the best 4 intervention targets from all the variables** which hopefully are the root causes of the other variables to start our analysis of their causal relations?
Let's think and analyze step by step. Then, provide your final answer (variable names only) within the tags <Answer>...</Answer>, separated by ", ".

### Insurance Warmup Prompt

You are a helpful assistant and expert in car insurance risks research. Here are some tips that you can pay attention to:
1. Assess whether there is a direct causal relationship, and consider potential confounding variables that might affect the relationship that could potentially not causal relationship.
2. Distinguish between correlations and causation; verify that correlations are not mistaken for causal relationships.

3. Ensure the correct temporal order of variables; confirm that the cause precedes the effect. Assuming we can do interventions to all the variables, your job is to assist in designing the best intervention experiments among the following variables to help discover their causal relations:

<ThisCarDam>: damage to the car
<MakeModel>: owning a sports car
<OtherCarCost>: cost of the other cars
<PropCost>: ratio of the cost for the two cars
<AntiTheft>: car has anti-theft
<DrivQuality>: driving quality
<DrivHist>: driving history
<MedCost>: cost of medical treatment
<Mileage>: how much mileage is on the car
<Antilock>: car has anti-lock
<CarValue>: value of the car
<Accident>: severity of the accident
<OtherCar>: being involved with other cars in the accident
<SeniorTrain>: received additional driving training
<ILiCost>: inspection cost
<SocioEcon>: socioeconomic status
<Theft>: theft occurred in the car
<Age>: age
<RuggedAuto>: ruggedness of the car
<GoodStudent>: being a good student driver
<VehicleYear>: year of vehicle
<HomeBase>: neighbourhood type
<ThisCarCost>: costs for the insured car
<Cushioning>: quality of cushioning in car
<RiskAversion>: being risk averse
<DrivingSkill>: driving skill
<Airbag>: car has an airbag

Assuming we can do interventions to all the variables, given the aforementioned variables and their descriptions, can you **echo your knowledge of those variables**, **temporally analyze** their relations, and then **choose the best 4 intervention targets from all the variables** which hopefully are the root causes of the other variables to start our analysis of their causal relations?

Let's think and analyze step by step. Then, provide your final answer (variable names only) within the tags <Answer>...</Answer>, separated by ", ".

---

### Alarm Warmup Prompt

You are a helpful assistant and expert in alarm message system for patient monitoring system research. Here are some tips that you can pay attention to:

1. Assess whether there is a direct causal relationship, and consider potential confounding variables that might affect the relationship that could potentially not causal relationship.

2. Distinguish between correlations and causation; verify that correlations are not mistaken for causal relationships.

3. Ensure the correct temporal order of variables; confirm that the cause precedes the effect. Assuming we can do interventions to all the variables, your job is to assist in designing the best intervention experiments among the following variables to help discover their causal relations:

<CATECHOL>: hormone made by the adrenal glands
<SAO2>: oxygen saturation of arterial blood
<VENTALV>: exchange of gas between the alveoli and the external environment
<ANAPHYLAXIS>: severe, life-threatening allergic reaction
<INSUFFANESTH>: whether there is insufficient anesthesia or not

---

<FIO2>: the concentration of oxygen in the gas mixture being inspired

<BP>: pressure of circulating blood against the walls of blood vessels

<PRESS>: breathing pressure

<VENTTUBE>: whether there is a breathing tube or not

<TPR>: amount of force exerted on circulating blood by vasculature of the body

<CO>: amount of blood pumped by the heart per minute

<PCWP>: pulmonary capillary wedge pressure

<ERRCAUTER>: whether there was an error during cautery or not

<KINKEDTUBE>: whether the chest tube is kinked or not

<PVSAT>: amount of oxygen bound to hemoglobin in the pulmonary artery

<INTUBATION>: process where a healthcare provider inserts a tube through a person's mouth or nose, then down into their trachea

<CVP>: measure of blood pressure in the vena cava

<HYPOVOLEMIA>: condition that occurs when your body loses fluid, like blood or water

<HRBP>: ratio of heart rate and blood pressure

<HREKG>: heart rate displayed on EKG monitor

<PAP>: blood pressure in the pulmonary artery

<EXPCO2>: expelled CO2

<ERRLOWOUTPUT>: error low output

<HISTORY>: previous medical history

<SHUNT>: hollow tube surgically placed in the brain (or occasionally in the spine) to help drain cerebrospinal fluid and redirect it to another location in the body where it can be reabsorbed

<VENTMACH>: the intensity level of a breathing machine

<VENTLUNG>: lung ventilation

<HRSAT>: measure of how much hemoglobin is currently bound to oxygen compared to how much hemoglobin remains unbound

<LVFAILURE>: occurs when there is dysfunction of the left ventricle causing insufficient delivery of blood to vital body organs

<DISCONNECT>: disconnection

<LVEDVOLUME>: amount of blood present in the left ventricle before contraction

<HR>: heart rate

<MINVOLSET>: the amount of time using a breathing machine

<PULMEMBOLUS>: sudden blockage in the pulmonary arteries, the blood vessels that send blood to your lungs

<STROKEVOLUME>: volume of blood pumped out of the left ventricle of the heart during each systolic cardiac contraction

<MINVOL>: amount of gas inhaled or exhaled from a person's lungs in one minute

<ARTCO2>: arterial carbon dioxide

Assuming we can do interventions to all the variables, given the aforementioned variables and their descriptions, can you **echo your knowledge of those variables**, **temporally analyze** their relations, and then **choose the best 4 intervention targets from all the variables** which hopefully are the root causes of the other variables to start our analysis of their causal relations?

Let's think and analyze step by step. Then, provide your final answer (variable names only) within the tags <Answer>...</Answer>, separated by ", ".

---

### LeGIT response for Fisram dataset

Let's start by echoing the variables and their descriptions:

1. **Spread**: Refers to the actual or potential spatial expansion across ecosystems, driven by climate, habitat, and transport.

2. **Estab**: Indicates the actual or potential for self-sustaining wild populations based on climate and habitat inputs.

3. **BehEff**: Represents the combined effect of predation and competition on native species behavior and viability.

4. **Clim**: Measures climate similarity scores to assess potential habitat suitability.

5. **HumEff**: Describes the combined influence of bites/toxins and other detrimental traits on humans.

6. **HabDis**: Capacity to modify or degrade habitat (erosion, eutrophication, sedimentation).

7. **EcoEff**: Overall impact of habitat disturbance, predation, and competition on ecosystem structure and function.

8. **Gen**: Capacity to affect native species' genetics via hybridization, GMO escape, or introgression.

9. **Harm**: Actual or potential physical or behavioral injury to native species and/or humans, or damage to habitats.

10. **Invas**: Final invasive-injurious outcome under the Lacey Act criteria, integrating Establishment, Spread, and Harm.

11. **Transp**: Combined human and non-human dispersal influence.

12. **Comp**: Capacity to compete with native species for food, space, or habitat.

13. **Pred**: Capacity to prey on and negatively affect native species populations.

14. **HabSuit**: Degree to which available habitat in the potential introduction area matches the species' known habitats.

15. **Path**: Role in spreading infectious agents to native wildlife.

16. **SppEff**: Overall impact of predation, competition, and genetics on native species viability.

17. **OthTr**: Non-bite/toxin traits posing human-health risks.

18. **NHumTrans**: Dispersal assistance by natural agents beyond the species' own movement.

19. **HumTrans**: Intentional or unintentional movement by humans.

20. **BiteEnv**: Direct adverse effects on human health via bites, stings, toxins, etc.

Now let's analyze the temporal relationships and potential causal links:

1. **Clim** and **HabSuit** are likely early indicators of whether a species can establish (Estab) and spread (Spread), as they relate to environmental suitability.

2. **Transp**, **HumTrans**, and **NHumTrans** are likely influential in the spread of species (Spread), as they involve movement and dispersal.

3. **Estab** leads to **Spread** as a species that can establish will likely expand its range.

4. **Spread** and **Estab** can contribute to **Invas**, as they are integral parts of invasive potential.

5. **BehEff**, **Comp**, and **Pred** are likely results of established populations interacting with native species.

6. **HabDis**, **EcoEff**, and **Harm** might be consequences of established and spreading populations.

7. **Gen** might affect native species' genetics once a species is established and interacting.

8. **Path** might be influenced by **Spread**, as wider distribution can lead to more interactions and potential for disease spread.

9. **HumEff**, **OthTr**, and **BiteEnv** are related to human impacts and may be less directly connected to ecological establishment and spread.

Considering the temporal order and potential causation:

- **Clim** and **HabSuit** are foundational for establishment and spread.
- **Transp** is crucial for movement and dispersal.
- **Estab** is a precursor to many other impacts.

Therefore, the best intervention targets to start analyzing causal relations are those that influence establishment and spread directly, as they are likely root causes:

<Answer>Clim, HabSuit, Transp, Estab</Answer>

---

**LeGIT response for Child dataset**

To address the problem effectively, we will follow a structured approach to understand the potential causal relationships among the variables. We will start by echoing the knowledge of

each variable, analyze their temporal relationships, and then identify the best intervention targets.

### Echoing Knowledge of Variables

1. **RUQO2**: Level of oxygen in the right upper quadriceps muscle.
2. **LVHreport**: Report of having left ventricular hypertrophy.
3. **BirthAsphyxia**: Lack of oxygen to the blood during the infant's birth.
4. **DuctFlow**: Blood flow across the ductus arteriosus.
5. **Disease**: Infant methemoglobinemia.
6. **LungParench**: The state of the blood vessels in the lungs.
7. **CO2Report**: A document reporting high levels of $CO_2$ in blood.
8. **LungFlow**: Low blood flow in the lungs.
9. **GruntingReport**: Report of infant grunting.
10. **Age**: Age of infant at disease presentation.
11. **LowerBodyO2**: Level of oxygen in the lower body.
12. **ChestXray**: Having a chest x-ray.
13. **CO2**: Level of $CO_2$ in the body.
14. **HypDistrib**: Low oxygen areas equally distributed around the body.
15. **Grunting**: Grunting in infants.
16. **XrayReport**: Lung excessively filled with blood.
17. **LVH**: Having left ventricular hypertrophy.
18. **CardiacMixing**: Mixing of oxygenated and deoxygenated blood.
19. **HypoxiaInO2**: Hypoxia when breathing oxygen.
20. **Sick**: Presence of an illness.

### Temporal Analysis

1. **BirthAsphyxia** is an early event that can lead to multiple downstream effects, such as hypoxia and potential cardiac issues.
2. **CardiacMixing** and **DuctFlow** are congenital or physiological conditions that can affect oxygenation and blood flow.
3. **HypoxiaInO2**, **LungFlow**, and **LowerBodyO2** are likely influenced by earlier conditions like **BirthAsphyxia**, **CardiacMixing**, and **DuctFlow**.
4. **CO2** and **CO2Report** are indicators of respiratory function and can be affected by lung and cardiac issues.
5. **LungParench**, **XrayReport**, and **ChestXray** are diagnostic or descriptive of lung conditions.
6. **Grunting** and **GruntingReport** are symptoms that may arise from respiratory distress.
7. **LVH** and **LVHreport** are indicative of cardiac stress or abnormalities, potentially secondary to other physiological issues.
8. **Disease** (infant methemoglobinemia) could be a result of hypoxic conditions and other metabolic issues.
9. **Sick** is a general indicator of illness and can be influenced by multiple preceding factors.

### Choosing Intervention Targets

To identify root causes, we need to target variables that likely influence many others and are early in the causal chain:

1. **BirthAsphyxia**: Early event potentially causing hypoxia, cardiac stress, and other downstream effects.
2. **CardiacMixing**: Affects oxygenation and can lead to hypoxia and other cardiac issues.
3. **DuctFlow**: Influences blood flow and oxygenation, impacting many other variables.
4. **LungFlow**: Directly affects oxygenation and can lead to hypoxia and respiratory distress.

These variables are likely to be root causes or early influencers in the causal network, making them ideal intervention targets to explore causal relationships.

### Final Answer

<Answer>BirthAsphyxia, CardiacMixing, DuctFlow, LungFlow</Answer>

**LeGIT response for Insurance dataset**

To analyze the causal relationships among these variables and select the best intervention targets, let's first categorize and understand the potential causal structure among them.
### Step 1: Echoing Knowledge of Variables
1. **<ThisCar>**: Costs for the insured car
2. **<Mileage>**: Mileage on the car
3. **<DrivQuality>**: Driving quality
4. **<RiskAversion>**: Being risk-averse
5. **<CarValue>**: Value of the car
6. **<MakeModel>**: Owning a sports car
7. **<ThisCarDam>**: Damage to the car
8. **<Theft>**: Theft occurred in the car
9. **<ILiCost>**: Inspection cost
10. **<Age>**: Age
11. **<OtherCar>**: Involvement with other cars in an accident
12. **<SeniorTrain>**: Received additional driving training
13. **<SocioEcon>**: Socioeconomic status
14. **<VehicleYear>**: Year of vehicle
15. **<HomeBase>**: Neighborhood type
16. **<DrivHist>**: Driving history
17. **<Accident>**: Severity of the accident
18. **<Airbag>**: Car has an airbag
19. **<AntiTheft>**: Car has anti-theft
20. **<ThisCarCost>**: Costs for the insured car
21. **<MedCost>**: Cost of medical treatment
22. **<Antilock>**: Car has anti-lock
23. **<GoodStudent>**: Being a good student driver
24. **<OtherCarCost>**: Cost of the other cars
25. **<Cushioning>**: Quality of cushioning in the car
26. **<RuggedAuto>**: Ruggedness of the car
27. **<PropCost>**: Ratio of the cost for the two cars
28. **<DrivingSkill>**: Driving skill
### Step 2: Temporal Analysis of Relations
- **Demographic and Personal Factors**:
- **<Age>**, **<RiskAversion>**, **<SocioEcon>**, **<GoodStudent>**: These are inherent traits or long-term conditions that can influence driving behavior and decisions.
- **Vehicle Characteristics**:
- **<VehicleYear>**, **<MakeModel>**, **<CarValue>**, **<Airbag>**, **<AntiTheft>**, **<Antilock>**, **<Cushioning>**, **<RuggedAuto>**: These are intrinsic to the car and can influence safety and risk but are typically decided before other events.
- **Driving Experience and Skills**:
- **<DrivQuality>**, **<DrivingSkill>**, **<DrivHist>**, **<SeniorTrain>**: These can evolve over time and influence accident likelihood.
- **Environmental Factors**:
- **<HomeBase>**: Affects exposure to theft and accidents.
- **Incident and Cost Factors**:
- **<ThisCarDam>**, **<Theft>**, **<Accident>**, **<ILiCost>**, **<ThisCarCost>**, **<MedCost>**, **<OtherCarCost>**, **<PropCost>**: These are outcomes or costs associated with incidents.
### Step 3: Choosing Intervention Targets
To identify root causes, we should consider intervening on variables that are likely to influence many others and are not themselves primarily effects of other variables. The best interventions would be on variables that are upstream in the causal chain and can potentially change multiple downstream outcomes.
1. **<RiskAversion>**: Can influence driving behavior, choice of car, and safety measures.
2. **<SocioEcon>**: Affects car choice, home base, and potentially driving habits.
3. **<MakeModel>**: Influences car value, safety features, and possibly driving style.

4. **<DrivingSkill>**: Directly impacts driving quality, accident likelihood, and associated costs.

These variables are chosen because they are fundamental characteristics or decisions that precede and potentially influence many other variables in the system.

### Final Answer

<Answer>RiskAversion, SocioEcon, MakeModel, DrivingSkill</Answer>

---

### LeGIT response for Insurance dataset

To determine the best intervention targets, we need to consider the relationships and possible temporal orders among the variables. Here's a step-by-step analysis:

### Step 1: Understanding the Variables

1. **Respiratory and Ventilation Variables:**
- **<VENTMACH>**: Intensity level of a breathing machine.
- **<MINVOLSET>**: Time using a breathing machine.
- **<VENTLUNG>**: Lung ventilation.
- **<PRESS>**: Breathing pressure.
- **<MINVOL>**: Minute volume; gas inhaled/exhaled per minute.
- **<VENTTUBE>**: Presence of a breathing tube.
- **<VENTALV>**: Gas exchange in alveoli.
- **<DISCONNECT>**: Disconnection.
- **<EXPCO2>**: Expelled CO2.
- **<ARTCO2>**: Arterial carbon dioxide.
- **<FIO2>**: Oxygen concentration in inspired gas.

2. **Cardiovascular Variables:**
- **<HREKG>**: Heart rate on EKG.
- **<PCWP>**: Pulmonary capillary wedge pressure.
- **<HRBP>**: Ratio of heart rate and blood pressure.
- **<BP>**: Blood pressure.
- **<CO>**: Cardiac output.
- **<HR>**: Heart rate.
- **<TPR>**: Total peripheral resistance.
- **<STROKEVOLUME>**: Blood volume per heart contraction.
- **<LVEDVOLUME>**: Blood in left ventricle before contraction.
- **<PAP>**: Pulmonary artery pressure.
- **<CVP>**: Central venous pressure.
- **<SAO2>**: Oxygen saturation of arterial blood.
- **<PVSAT>**: Oxygen saturation in pulmonary artery.
- **<HRSAT>**: Hemoglobin oxygen saturation.
- **<CATECHOL>**: Adrenal hormones.

3. **Medical Conditions and Procedures:**
- **<ANAPHYLAXIS>**: Severe allergic reaction.
- **<INTUBATION>**: Insertion of a breathing tube.
- **<INSUFFANESTH>**: Insufficient anesthesia.
- **<PULMEMBOLUS>**: Pulmonary embolism.
- **<HYPOVOLEMIA>**: Fluid loss condition.
- **<LVFAILURE>**: Left ventricle failure.
- **<SHUNT>**: Drainage tube for cerebrospinal fluid.
- **<KINKEDTUBE>**: Kinked chest tube.

4. **Errors and Anomalies:**
- **<ERRLOWOUTPUT>**: Low output error.
- **<ERRCAUTER>**: Cautery error.

5. **Other:** - **<HISTORY>**: Previous medical history.

### Step 2: Temporal Analysis

- **<INTUBATION>** and **<VENTMACH>** are likely initial interventions that can influence many respiratory and cardiovascular variables.

- **<ANAPHYLAXIS>**, **<PULMEMBOLUS>**, and **<HYPOVOLEMIA>** are acute conditions that can cause significant changes in cardiovascular and respiratory variables.
- **<INSUFFANESTH>** can affect heart rate and blood pressure due to inadequate sedation.
- **<VENTLUNG>**, **<VENTALV>**, and **<MINVOL>** are outcomes of ventilation interventions.

### Step 3: Choosing the Best Intervention Targets

To determine causality effectively, we want to target variables that are likely root causes or significant influencers:

1. **<INTUBATION>**: This is a direct intervention that can affect many respiratory variables.
2. **<VENTMACH>**: Controls the intensity of mechanical ventilation, influencing respiratory and possibly cardiovascular variables.
3. **<ANAPHYLAXIS>**: A condition that can have widespread effects on cardiovascular and respiratory systems.
4. **<HYPOVOLEMIA>**: Can significantly impact cardiovascular variables like blood pressure and cardiac output.

These interventions can provide insights into the causal relationships within the system.

### Final Answer

<Answer>INTUBATION, VENTMACH, ANAPHYLAXIS, HYPOVOLEMIA</Answer>

