# OpenReview forum: "LeGIT: LLM Guided Intervention Targeting for Online Causal Discovery"
_ICLR.cc/2026/Conference — Submitted to ICLR 2026_

### Official Review · Reviewer_AwPk · 2025-10-28

**Soundness:** 2
**Presentation:** 2
**Contribution:** 2
**Rating:** 2
**Confidence:** 4

**Summary:**

The authors propose a novel hybrid framework for causal structure learning that combines the vast prior world knowledge of LLMs with existing methods (referred to as numerical methods). The method consists of four stages: (i) Warmup Stage annd (ii) Bootstrapping Stage where LLMs are employed, following by a (iii) Re-Sampling Stage and (iv) Numerical-Method Stage** where the search is continued with “traditional” causal discovery methods. Eventually, the authors assess the proposed method on four datasets of the BN benchmark suite, and compare the performance to three online causal discovery methods, as well 3 heuristics and a human reference score.

**Strengths:**

- The idea to leverage the vast world-knowledge of LLMs as a prior for causal discovery methods is definitely an interesting idea that deserves more exploration.
- Clean Visualizations

**Weaknesses:**

- **Clarity:** The paper is somewhat hard to parse. Despite reading section 3.2 multiple times, I still find it hard to understand the proposed method in detail and cannot confidently say that I am 100% understanding what's going on.
- **Variance:** The variances in table 1 seem to be huge. When considering the standard deviation over 5 seeds, GIT and LeGIT seems to be overlapping in most cases, making it hard to make a conclusion about the effectiveness of the proposed approach.
- **Contamination:** While the author address the issue of contamination (lines 100 to 108), they refer to Long et al. (2023) as an evidence that prominent LLMs struggle to reconstruct causal graphs. However, the referred work tested GPT3 while the present work builds upon OpenAI's 4o. LLMs have come along way since GPT3, and it's very likely that results would look different with today's state of the art models, especially reasoning models. Hence, the effect of contamination cannot really be judged given the presented evidence. Instead of asking for interventional targets, it would be worthwhile to check which edges the LLMs are able to identify?
- **Theoretical Argument:** The theoretical argument is somewhat superfluous -- could you please provide more depth on what the theoretical argument provides? The entire theoretical novelty hinges on empirically verifying Assumption D.2. If the authors cannot show the LLM targets consistently satisfy this property, the whole theoretical arguments becomes a tautology with inherited proof from other work.

**Questions:**

- Could you please provided a clearer description on the algorithm?
- Have the authors also experimented to initialize the graph with LLM proposed edges instead of using the LLM to suggest informative intervention nodes?
- Have the authors experimented with different prompts in the warmup stage? I suspect that the LLM's performance on discovering useful variables may largely vary between prompts. And the given prompt seems to be slightly ambigious -- so I would be curious how well the model performs with a more precise prompt.
- Could the authosr please run LLM baselines where the model directly predicts causal edges among given variables.
-  Line 86: "Relatively clearer" --> what do you mean with this?

**Details Of Ethics Concerns:**

-

---

> ### Author Response · Authors · 2025-12-02
> **Response to Reviewer AwPk (Part 1)**
>
> Dear Reviewer AwPk:
>
> We appreciate your time and effort in reviewing our work. Below we provide detailed responses to your concerns and questions.
>
> >W1 & Q1: Clarity of the method
>
> We apologize if the description was hard to parse.  Here we provide a clear description:
> - Step 1: Warmup (LLM-driven): At $t=0$, numerical methods have zero data and guess randomly. We query the LLM: "Given these variable descriptions, which are likely root causes?" We intervene on these proposed targets to collect the first batches of data.
>
> - Step 2: Bootstrap (Feedback Loop): After a few rounds, we have a preliminary graph structure (often fragmented). We identify nodes that are currently isolated or unvisited. We feed this list back to the LLM: "Focus on these specific variables; are there important causes we missed?" This ensures coverage of the graph.
>
> - Step 3: Re-sampling (Verification): We strictly intervene on the LLM-proposed targets one more time. Doubling down ensures the numerical method receives a strong enough signal.
>
> - Step 4: Numerical (Data-driven): Once the "cold start" phase is over and we have a decent graph skeleton, we switch entirely to the numerical method (GIT/CBED) to perform fine-grained optimization, as data is now sufficient for it to work effectively.
>
> >W2: Variance
>
>  To prove this is not noise, we conducted Paired T-tests and Wilcoxon Signed-Rank tests on 10 seeds. As reported in Appendix C.4 (Table 6 & 7), **the p-values for LeGIT vs. GIT are consistently < 0.05 (e.g., 0.0172 for Alarm SHD). This statistically confirms that LeGIT's improvement is real and robust, not an artifact of variance**.
>
> >W3: Contamination
>
> Observing the structure and generating accurate intervention targets are fundamentally different tasks. There is very limited documentation on how the causal graphs used in our experiments were originally obtained via interventional experiments, so it is unlikely that LLMs have been trained on, or can directly memorize, the intervention selection process itself.
>
> **Importantly, Jiralerspong et al. (2024) and Khatibi et al. (2024) use GPT-4–class models to directly predict edges from variable descriptions on exactly the kind of benchmarks we study**. These works show that even with access to rich metadata, GPT-4–level models do not recover the correct structures for several datasets we use (e.g., Alarm/Child), even if they may have seen related content during pretraining. This suggests that current LLMs are not simply memorizing and reproducing the ground-truth graphs.
>
> To further address the concern that GPT-4o might just “know” these specific datasets, we also evaluated LeGIT with open-source models such as Llama-3.1-405B and DeepSeek-V3 (Appendix C.3). **These models are trained on different data and likely have different contamination profiles, yet LeGIT consistently maintained its performance advantage across them**. This cross-model robustness makes a “memorized benchmark” explanation less plausible.
>
> Finally, we believe hard-coding LLM-predicted edges as priors is inherently risky: if the LLM hallucinates a non-existent edge, it injects a strong structural bias that is difficult for the numerical method to unlearn. In contrast, LeGIT uses the LLM only to select interventions, i.e., to decide which data to collect. **This keeps the influence of the LLM “soft”: if the LLM is wrong about a relation, the resulting interventional data will simply not support that edge, and the numerical method will correctly discard it**. This design makes LeGIT significantly more robust to contamination and hallucination than approaches that directly trust LLM-predicted edges.
>
> >W4: Theory part
>
> The theoretical section is intended to explain why the hybrid approach works, formally linking the "cold start" problem to convergence rates.

---

> ### Author Response · Authors · 2025-12-02
> **Response to Reviewer AwPk (Part 2)**
>
> >Q2: Experimented with initializing the graph with LLM edges instead of targets?
>
> Yes, we did consider using the LLM to initialize the graph directly (e.g., by asking it to propose edges) instead of using it only for intervention targets. However, we decided against this design in the current work for two main reasons:
>
> 1.	API cost and complexity. Methods that retrieve edges via LLMs (such as pairwise querying or BSF-style procedures) typically require many more LLM calls than our current design. They query the model for each ordered pair (or subset) of variables, which quickly becomes expensive and slow as the number of variables grows. **In contrast, LeGIT calls the LLM only a small, fixed number of times to obtain warmup and bootstrapped target sets, making it much more cost- and time-efficient**.
>
> 2.	Limited benefit from initialization for ENCO. Our backbone method, ENCO, is relatively insensitive to the initial graph: after a few intervention rounds, the posterior over structures is dominated by the accumulated observational + interventional data rather than the starting point. In this regime, using LLMs to set the initial edges brings less benefit than using them to decide where to intervene early on, which directly affects what data are collected. For this reason, we chose to focus the LLM’s capacity on selecting informative intervention targets instead of trying to pre-specify the entire edge set.
>
> We see LLM-based graph initialization as complementary to our approach, but given the cost profile and ENCO’s weak dependence on initialization, using LLMs for target selection is the more impactful and scalable choice in our current framework.
>
> >Q3: Prompt
>
> We agree that prompt design is an important factor. In this submission, we use a single prompt template (Fig. 3 + dataset-specific instantiations in Appendix E) across all datasets and LLMs. And the only dataset-specific content is the list of variable names and short descriptions; we do not hand-tune wording per dataset or per LLM. We also apply a standard “self-consistency” trick, but otherwise keep prompting simple on purpose, to avoid extensive hand-engineering.
>
> >Q4: Could the author please run LLM baselines where the model directly predicts causal edges among given variables?
>
> We cite works that do this (e.g., Jiralerspong et al., 2024). Pure LLM edge prediction is a static task. LeGIT solves the Online task (Active Learning). Comparing them directly is a category error: LeGIT generates data to learn a graph, whereas direct prediction generates a graph without data (or with frozen data). LeGIT is designed to verify and refine graphs using data, which pure LLM prediction cannot do.
>
> >Q5: Line 86: "Relatively clearer" --> what do you mean with this?
>
> Our intent was to express that:
> - In the very early rounds, **the estimated graph/posterior is highly uncertain and unstable; there, global meta-knowledge (from an LLM or an expert) can help avoid very poor initial interventions**.
> - After sufficient interventions and updates, ENCO’s graph posterior becomes partially stabilized: many high-degree nodes and major edges acquire high confidence. At this stage, the marginal benefit of additional high-level semantic knowledge is smaller, and fine-grained numerical updates are more reliable.

---

### Official Review · Reviewer_dDFN · 2025-10-31

**Soundness:** 1
**Presentation:** 3
**Contribution:** 2
**Rating:** 2
**Confidence:** 4

**Summary:**

The paper introduces LeGIT, an intervention-targeting method for experimental design in online causal discovery. LeGIT leverages the meta-knowledge embedded in large language models (LLMs) to identify which nodes should be selected for intervention. The LLM-enhanced framework can be integrated with any causal discovery algorithm that utilizes interventional data. In this setup, the LLM receives a prompt summarizing the metadata of the graph’s vertices (e.g., variable descriptions) and responds with a set of proposed interventional nodes based on its internal and contextual knowledge. The framework further incorporates bootstrap and resampling stages to enhance performance and is compared against alternative intervention-selection algorithms that do not exploit such meta-knowledge.

**Strengths:**

The topic addressed in the paper is highly engaging, as it pushes the boundaries of intervention targeting in causal discovery by integrating external meta-knowledge from large language models (LLMs). Exploring the applicability of LLMs to this problem offers an insightful avenue for assessing their reasoning abilities and the extent of their embedded common and expert knowledge. The idea of treating an LLM as an artificial expert in the absence of a human counterpart is particularly compelling, making the comparison between LLM-selected and human-selected intervention targets especially valuable.
I particularly appreciate the experiments presented in Figure 2, which provide intuition on how the set of intervention vertices may vary across methods, and Table 2, which—especially when compared to Table 1—illustrates how LeGIT’s performance is influenced by changes in the data budget. However, I would strongly encourage the inclusion of additional experiments analyzing how the proposed enhancements of LeGIT affect overall performance (see the discussion on soundness in the weaknesses section).

**Weaknesses:**

**Notes on Soundness:**

My most serious concern is that the comparison between the intervention selection algorithms and the LLM-enhanced algorithm is not entirely fair. This is because the LLM has access to additional metadata related to the studied graph, leveraging knowledge that the other algorithms were never designed to use (except for the input provided by human experts). In other words, the assumptions regarding the input data for the causal discovery problem have been extended. It is therefore quite expected that incorporating this extra knowledge—on top of the standard framework—would lead to better results. This is not to say that the impact of such an LLM-based enhancement is uninteresting to study; however, it is important to recognize that LeGIT addresses a different problem than, for example, GIT, AIT, or CBES. This distinction, however, is not reflected in the narrative presented by the authors.

I would suggest that the authors focus on comparing their updated Online Causal Discovery framework with other LLM-guided causal discovery algorithms. For instance, the works of Jiralerspong et al. and Khatibi et al. rely on similar access to metadata from LLMs to uncover causal dependencies, and they use the same or similar graphs and metrics (other comparable algorithms are also summarized in Wan et al.). The experiments with GIT, AIT, and CBES serve more as a teaser or example of LeGIT’s potential, rather than as a fair comparison between methods that have access to the same data.

The authors could therefore also adjust their narrative to focus on understanding the impact of incorporating this additional knowledge. For example, which nodes are most affected? (Figure 2 is a nice example, although it might be more informative to compare against a simpler case where the ideal distribution can be directly derived. In the graphs from Figure 2, it is difficult to assess which intervention set is correct, or whether the interventions occur on edges that can be reversed within a MEC class—and hence whether the intervention is actually required to achieve identifiability.) Similarly, how do the different enhancements (bootstrapping, resampling) affect convergence or final performance? Does including information about the current graph in the prompt improve or deteriorate performance? How does performance change with the number of nodes proposed by the LLM—and can the LLM propose a ranking of nodes? These are just some examples of how the authors could build a stronger narrative around the ablation studies and deepen the understanding of their framework  while staying within the same online causal discovery framework.

**Notes on Experiments Setup:**

Throughout the paper, the authors use only four graphs of more or less similar size. This is, of course, a reasonable starting point, but I wonder why these specific graphs were selected. More importantly, I see no clear restriction preventing the extension of the evaluation to other networks (e.g., additional graphs from https://www.bnlearn.com/bnrepository/
Moreover, based on the description of LeGIT, it appears that the method can be used with any algorithm $A$ that satisfies the conditions for online causal discovery (i.e., potentially not even gradient-based). Did the authors attempt to use different base causal discovery algorithms besides ENCO? Even if not as a main experiment, such a demonstration could nicely illustrate the universality of the proposed approach.

**Notes on Contribution:**

I find the topic quite interesting, as it explores the integration of LLMs’ common meta-knowledge—akin to the bias present in human experts—into causal discovery. However, this perspective has also been explored in prior work (e.g. Jiralerspong et al., Khatibi et al., Wan et al.). While focusing on scoring the intervention potential of each node appears novel, the experimental design (see “Soundness” above) does not give me enough confidence to fully assess the strength of the contribution.

**Notes on presentation (Minor):**

The paper is generally well written, but I do think that some of the sections could use more attention. For instance, Preliminaries  do not explain all the notations in Algorithm 1 (e.g. what is $P(G)$), there is a discrepancy between $\phi$ in  Algorithm 1 (in “Output”) and in text (line 133). It is not explained what those parameters are and how $A$ updates them (in practice we also have functional parameters that are also updated by the algorithm A and can thus influence the structural ones - see Lippe et al.). In Algorithm 2, line 8, should it not be $D_{bootrstapped}[i-T_{warmup}]$?

**References:**

Jiralerspong, Thomas, et al. "Efficient causal graph discovery using large language models." arXiv preprint arXiv:2402.01207 (2024).
Khatibi, Elahe, et al. "Alcm: Autonomous llm-augmented causal discovery framework." arXiv preprint arXiv:2405.01744 (2024).
Wan, Guangya, et al. "Large Language Models for Causal Discovery: Current Landscape and Future Directions." arXiv preprint arXiv:2402.11068 (2024).
Lippe, Phillip, Taco Cohen, and Efstratios Gavves. "Efficient neural causal discovery without acyclicity constraints." arXiv preprint arXiv:2107.10483 (2021).

**Questions:**

The current algorithm primarily relies on meta-knowledge derived from the descriptions of the variables. I was wondering whether the authors also explored incorporating contextual information from the graph itself (e.g., details about the MEC class) or integrating principles from do-calculus or v-structure verification. Could such approaches work even in cases where no node descriptions are provided? In other words, could the LLM base its reasoning solely on causal discovery knowledge and the graph structure to refine the current solution and subsequently select the intervention nodes? In that way LeGiT could be used even on fully artificial graphs, or models when no previous bias/knowledge is available about the nodes.

In Figure 4, it appears—especially for the Child and Fisram graphs—that LeGIT does not improve during the early acquisition steps and then sharply drops. Could this breaking point correspond to when the numerical scoring algorithms start to take effect?

In Algorithm 2, the LeGIT intervention targets are selected at the beginning of training. Did the authors examine whether incorporating them as the final interventions (or interleaving them with the numerically scored ones) provides any benefit?

I do generally believe it could be an interesting paper, but currently it suffers from its positioning (i.e. the algorithm works well, but it is either not fairly comparable to other approaches - beacuse of the additional knowledge - or other LLM-guided causal discovery approaches could be adapt to work within the online causal discovery framework to allow for a comparision with meta-data).

---

> ### Author Response · Authors · 2025-12-02
> **Response to Reviewer dDFN (Part 1)**
>
> Dear Reviewer dDFN:
>
> Thank you for your time and insightful comments on our work. Below, we provide detailed responses to your questions.
>
> > W1: Notes on Soundness:
>
> We fully acknowledge that LeGIT utilizes extra information (variable descriptions) that standard numerical baselines do not. However, we respectfully argue that this is the core contribution rather than a flaw in fairness. Rather, it is an emerging and new direction opened by LLMs [1-3].
>
> Reframing the Problem: The "standard framework" of ignoring available metadata is precisely the limitation we aim to overcome. In almost all real-world scientific discoveries (biology, economics), metadata exists. Ignoring it to satisfy a purely numerical problem definition is artificial. **LeGIT aims to quantify how much this readily available information can accelerate discovery**.
>
> The "Fair" Baseline: The most "fair" comparison in our paper is the Human Expert baseline. Humans do use the same metadata and reasoning capabilities as the LLM. The fact that LeGIT matches or outperforms Humans (Table 1) validates its soundness as an automated "Artificial Expert," regardless of the numerical baselines.
> Existing works like Jiralerspong et al. and Khatibi et al. primarily focus on static causal discovery (inferring the graph once from data/text). LeGIT focuses on Online Active Learning (sequentially selecting interventions to generate new data). These are fundamentally different tasks.
> Our focus is on the intervention targeting policy, which is why we compared against intervention strategies (AIT/GIT). As the availability of additional intervention data and the incorporation of a theoretically guaranteed data-driven causal discovery method, our approach unfairly outperforms the results reported in Jiralespong et al. and Khatibi et al. on the same dataset (Child).
>
> > W2.1: Notes on Experiment Setup Dataset
>
>  We chose Alarm, Insurance, Child and Fisram because:
> They are standard benchmarks from the BN repository with widely used ground-truth DAGs.
> Crucially for our setting, they come with meaningful variable names and descriptions, enabling non-trivial meta-knowledge for both LLMs and human experts. Many other bnlearn networks either lack such descriptions or have extremely short labels, which limits the role of meta-information.
> They cover reasonably diverse domains (medical monitoring, car insurance, pediatrics, and industrial system).
>
> Following your suggestion, we run an experiment on Asia (8 node) dataset same setting as Table 1 to examine the generalizability of LeGIT.
>
> | Method        | SHD  | SID  | BSF    |
> |---------------|------|------|--------|
> | CBED          | 2.20 | 4.60 | 0.7833 |
> | AIT           | 1.80 | 6.60 | 0.7833 |
> | Random Choice | 1.20 | 2.20 | 0.8708 |
> | Round Robin   | 1.40 | 2.20 | 0.8250 |
> | Degree Prob   | 1.00 | 1.00 | 0.8750 |
> | GIT           | 1.00 | 1.00 | 0.8750 |
> | Human         | 1.40 | 3.60 | 0.8667 |
> | LeGIT         | 0.80 | 0.80 | 0.9000 |
>
> > W2.2: different base causal discovery algorithms
>
> While most experiments use ENCO+GIT as the base numerical method, we already include a CBED pairing experiment in Table 3, showing that LeGIT also improves CBED’s SHD/SID on Alarm and Child with the same budget. Also, we have additionally add small-scale   with DCDI backbone on ASIA dataset, the results are as follow(Detailed Setting please refer to our response to Reviewer i43n) :
>
> | Method      | MeanSHD |
> |------------|---------|
> | RoundRoubin | 10.2   |
> | CBED       | 9.8     |
> | LeGIT+CBED | 8.4     |
> | GIT        | 8.6     |
> | LeGIT      | 8       |
>
> From the results, we can find that, LeGIT still posits advantages over previous methods with other active causal discovery algorithms as the backbone.

---

> ### Author Response · Authors · 2025-12-02
> **Response to Reviewer dDFN (Part 2)**
>
> >Q1: explored incorporating contextual information from the graph itself
>
> This is a very insightful suggestion. We did attempt to provide the LLM with additional information (MEC class) in the early stages of this project, but found that the large model's performance was unstable on complex structures, resulting in insufficient robustness. This is consistent with the findings of  Chen, S et al. [4]  Sheth I et al. [5]. Therefore, at this stage, we intentionally restrict the LLM's access to: variable names, short descriptions, and high-level domain context, and require it to infer the root cause, thereby achieving more robust performance. We assign tasks to the component best suited for them. LLMs are excellent at Semantics (Variable A causes B), while Numerical Methods (GIT/ENCO) are excellent at Structure/Statistics (MEC, consistency). Relying on LLMs for structural reasoning on artificial graphs (without semantic clues) would likely degrade performance to random guessing.
>
> > Q2: Figure 4 "Breaking Point" - does this correspond to the numerical switch?
>
> Yes, the more pronounced drop you noticed for Child and Fisram in Figure 4 happens roughly around the handover between the LLM-based first 10 rounds and the subsequent numeric stage.   **This is consistent with our intuition: once enough informative interventions on those influential nodes have been collected, the numerical method (GIT) can use that “better initialization” to accelerate the discovery process**.
>
> >Q3: Why select targets at the start? Did you try interleaving?
>
> We focus on the start because that is where numerical methods fail most catastrophically (due to high variance in gradient estimators with $N \approx 0$). And we find that fully interleaving at late stages yields diminishing returns because the numerical method eventually accumulates enough data to surpass the static semantic knowledge of the LLM.
>
>
> Reference：
>
> [1] Liu, Chenxi, et al. "Discovery of the hidden world with large language models." Advances in Neural Information Processing Systems 37 (2024): 102307-102365.
>
> [2] Abdulaal, Ahmed, et al. “Causal Modelling Agents: Causal Graph Discovery through Synergising Metadata- and Data-Driven Reasoning.” The Twelfth International Conference on Learning Representations, 2024, https://openreview.net/forum?id=pAoqRlTBtY.
>
> [3] Vashishtha, Aniket, et al. Causal Order: The Key to Leveraging Imperfect Experts in Causal Inference. The Thirteenth International Conference on Learning Representations, 2025, https://openreview.net/forum?id=9juyeCqL0u.
>
> [4] Chen, S. et al. 2024. CLEAR: Can Language Models Really Understand Causal Graphs? In Findings of the Association for Computational Linguistics: EMNLP 2024. Miami, Florida, USA. Association for Computational Linguistics.
>
> [5] Sheth, I. et al. 2025. CausalGraph2LLM: Evaluating LLMs for Causal Queries. In Findings of the Association for Computational Linguistics: NAACL 2025. Albuquerque, New Mexico, USA. Association for Computational Linguistics.

---

### Official Review · Reviewer_tK3A · 2025-11-03

**Soundness:** 3
**Presentation:** 3
**Contribution:** 3
**Rating:** 8
**Confidence:** 3

**Summary:**

This paper proposes a framework to incorporate LLMs into existing gradient-based intervention targeting methods. Specifically, the paper proposes using LLMs in a warmup phase that includes an initial guess for the causal graph based on prompting the LLM with the variable names in the dataset.  The paper performs ablations and shows that on 4 datasets, LeGIT, using GIT as underlying gradient-based intervention targeting method, outperforms vanilla GIT as well as a version of LeGIT where the LLM was replaced by human feedback.

**Strengths:**

Presentation of work is clear and results seem positive; the idea of using LLMs to resolve the issue of instability under certain initializations for gradient-based intervention targeting methods seems novel to me.

**Weaknesses:**

Experiments on only four standard datasets could be too few to draw statistically significant conclusions, especially if any optimizations (e.g. tuning prompts; tuning length of the warmup phase) have been performed using these datasets. It would be good to see robustness to cases where datasets are less well-known and LLMs may have difficulty coming up with a reasonable guess of the underlying structure.

**Questions:**

- Experiments have been performed iterating over 5 seeds. Are 5 seeds sufficient to obtain statistically significant results?
- How was tuning the length of the warmup relative to the length of the experiments done? Are any plots showing performance under varying warmup-length?
- Is there a possibility to experiment on more novel datasets where guessing the causal graph may be difficult for an LLM? It would be interesting to see to what extent LeGIT still outperforms or whether incorporating LLMs still helps in scenarios that are closer to "unsolved" cases, where there is much less background knowledge available to the LLM; as this would be relevant for real research problems.

---

> ### Author Response · Authors · 2025-12-02
> **Response to Reviewer tK3A**
>
> Dear Reviewer tK3A:
>
> We thank the reviewer for their positive assessment (Score: 8) and for recognizing the novelty of using LLMs to resolve initialization instability in gradient-based intervention targeting. Below we address the weaknesses and questions raised.
>
> > W1: Experiment on four datasets
>
> We agree that robustness beyond a small set of benchmarks is important, and we will clarify in the paper how we tried to avoid overfitting:
> **No per-dataset prompt tuning**. All experiments use the same prompt template (Fig. 3 + Appendix E) across datasets. The only dataset-specific content is the list of variable names and their short descriptions; we did not manually adjust wording or instructions per dataset to optimize performance.
> **Fixed warmup schedule across datasets**. We use a single global setting $T_{warmup} = 3$, $T_{\text{bootstrapped}} = 2$ for all experiments (Alarm, Insurance, Child, Fisram), with total budget T = 33. These values were chosen as a simple, small fraction of the overall budget so that the LLM-guided phase is short and the numerical backbone dominates later rounds; we did not tune them per dataset or per model. We will state this explicitly in Section 4.1 to make clear that LeGIT is not tailored to individual graphs.
> **Diverse domains within the four benchmarks**. While all four graphs come from the BN repository, they span distinct domains and vocabularies (medical monitoring, insurance, pediatric health, and a relatively niche ecological risk model for invasive fish species). In particular, Fisram uses specialized terminology that is much less likely to appear frequently in LLM pre-training data, and LeGIT still improves over GIT and the human baseline there.
>
> We fully share your interest in a broader evaluation. Due to space and available benchmark constraints we focused on quality of analysis on these four benchmarks, but we view LeGIT as a framework and will emphasize in the conclusion that extending to more novel / anonymized datasets is a key next step.
>
>
> > Q1: Are 5 seeds sufficient to obtain statistically significant results?
>
> Yes, while 5 seeds are standard in this domain[1], we have verified our results with 10 seeds for the Alarm and Child datasets to ensure statistical validity. As detailed in Appendix C.4, LeGIT consistently outperforms the baselines with 10 seeds. **We performed paired t-tests and Wilcoxon signed-rank tests, obtaining p-values well below 0.05, confirming that the improvements are statistically significant and not due to random variance**.
>
> > Q2: How was tuning the length of the warmup relative to the length of the experiments done?
>
> We set the warmup length ($T_{warmup}=3$) as a small fixed fraction ($\approx 10$%) of the total experimental budget ($T=33$). We did not extensively tune this hyperparameter per dataset to demonstrate the generalizability of the method. The goal was to provide just enough initial "priors" to steer the numerical method away from poor local optima without consuming the entire budget.
>
> > Q3: Is there a possibility to experiment on more novel datasets where guessing the causal graph may be difficult for an LLM?
>
> This is an excellent direction and closely relates to the limitation you point out. Our current submission takes a first step. Even though three of the four graphs are classical BNs (Alarm, Insurance, Child), **the Fisram graph is from a specialized ecological risk-assessment domain with non-everyday terminology (e.g., invasive species risk factors)**. LeGIT still provides gains over GIT and human baselines there, suggesting that it can leverage partial domain knowledge even when the setting is not widely known.
>
> Currently, due to limitations in real-world Baysian networks, most of the graphs described are meaningless (e.g., X1, X2), making it impossible to test larger and more complex graphs. However, we need to clarify that the adopted causal graphs are already among the largest compared to other works using LLMs for causal discovery, as well as those used in numerical methods for online causal discovery.
>
>
> Reference: Lippe P, Cohen T, Gavves E. Efficient neural causal discovery without acyclicity constraints[J]. arXiv preprint arXiv:2107.10483, 2021.

---

### Official Review · Reviewer_i43n · 2025-11-03

**Soundness:** 1
**Presentation:** 3
**Contribution:** 1
**Rating:** 2
**Confidence:** 3

**Summary:**

The work explores the usage of LLMs for experimental design. The authors propose a framework and prompts for acquiring information about root causes in the network from LLMs. This information guides further experimentation. The approach is compared to several baselines on four real-world inspired problems using the ENCO algorithm to obtain structure posteriors.

**Strengths:**

1. The paper touches on an important problem of efficient experimental design.
2. The paper is easy to read.

**Weaknesses:**

1. The method description is unclear. The “Re-sampling stage” paragraph states that each LLM suggestion undergoes filtering (must “survive” two independent votes). Meanwhile, in Algorithm 2:
* In line 4, we intervene on all variables from the  warmup list
* In line 8, we intervene on all variables from the bootstrap-warmup list
* In line 12, we intervene on all variables from both datasets once again

Please clarify this discrepancy in descriptions.
Also, in the bootstrap stage, how is the intermediate causal discovery result leveraged?

2. The proposed method relies on LLMs identifying root causes. What is the theoretical motivation for including such a suggestion in the prompts? In general, selecting optimal experimental designs is a more complex task [Eberhardt].

3. The observed effectiveness of the method may relate to how the ENCO method works and may fail to generalize. Note that ENCO has no convergence guarantees when the number of interventions is less than d-1. The ENCO framework only guarantees the recovery of the correct graph when interventions were conducted on at least n-1 nodes. It would be beneficial for the applicability (and possibly performance) to use a method with stronger theoretical guarantees under partial intervention sets, for example, DCDI or DIBS.

4. The approach relies on the ability of LLMs to reason about the variables. This ability can be impaired in at least two cases: insufficient information about variables (when the descriptions are missing or are non-informative), lack of background knowledge. The discussion on these two limitations is missing from the paper. When applying this method to a new setting, how can I make sure both requirements are fulfilled and trust in the effectiveness of the method? The work also lacks discussion about the effectiveness of the proposed prompt and the consistency of LLMs' responses.

5. There are no experiments or analyses that evaluate the scalability and cost efficiency of the proposed method to support he claim “LLMs offer a scalable and cost-efficient approach to enhance experimental design” (line 485). Moreover, since LeGIT requires using a numerical intervention targeting method, the cost efficiency and scalability seem to be similar to existing methods.

**References**

[Eberhardt] Eberhardt, Almost Optimal Intervention Sets for Causal Discovery

[DCDI] Brouillard et al., Differentiable Causal Discovery from Interventional Data

[DIBS] Lorch et al., DiBS: Differentiable Bayesian Structure Learning

**Questions:**

1. What is the prompt for the boostraped-warmup intervention set acquisition?
2. Why are the results on statistical significance, in section 5.2, partial? How do the test results look for other considered graphs?
3. Line 468: “rapid interventions are required” - Can the Authors provide an example of a real-world problem where rapid experimentation is needed and is costly?
4. Line 93 typo: intervening -> intervention

---

> ### Author Response · Authors · 2025-12-02
> **Response to Reviewer i43n (Part 1)**
>
> Dear Reviewer i43n:
>
> Thanks for your comment. Below, we address the specific concerns and questions raised.
>
> > W1.1 Method Description
>
> We apologize for the confusion caused by the phrasing in the "Re-sampling stage" paragraph regarding "surviving two independent votes."  As shown in Algorithm 2 (Lines 4, 8, and 12), **we indeed intervene on the variables suggested by the LLM. By revisiting these targets (re-sampling)**, we ensure that the numerical algorithm receives sufficient statistical evidence to either confirm or reject the structural hypotheses suggested by the LLM.
>
> > W 1.2 Intermediate causal discovery result leveraged in the bootstrap stage
>
> As outlined in Algorithm 2 (Line 6: $V_{unvisited}$ <- Unvisited nodes from P(Gi)), we use the intermediate graph structure learned after the initial warmup rounds to identify nodes that have not yet been targeted or are part of unresolved sub-graphs (e.g., isolated nodes or unconnected components). We feed this list of $V_{unvisited}$ (variables describing the "leftover" parts of the system) back into the LLM prompt. This guides the LLM to focus its reasoning specifically on these unexplored areas, rather than redundantly suggesting the same root causes identified in the first stage.
>
> > W2 Theoretical motivation for including the suggestion on identifying root causes in the prompts
>
> We agree that optimal experimental design is a complex task often requiring orientation of specific edges rather than just finding root causes. LeGIT uses "root cause identification" as a heuristic prior, not a replacement for optimal design theories. **In the "zero-shot" phase of online discovery (where $N=0$ interventions), numerical methods like GIT often behave like random guessers because they lack the data to compute gradients or information gain meaningfully.  Intervening on root causes (upstream nodes) is information-theoretically highly valuable in the early stages because it propagates effects to all descendants**, maximizing the number of variables perturbed. This helps quickly establish a "skeleton" of the graph (Theorem 3.1 and Appendix B in ENCO paper). Once this skeleton is established, LeGIT hands over control to the numerical method (e.g., GIT) to perform the fine-grained.
>
> > W3 Generaliability and applicability (and possibly performance) to use other methods with stronger theoretical guarantees under partial intervention sets, for example, DCDI or DIBS.
>
> Conceptually, LeGIT is backbone-agnostic: the LLM-guided warmup only decides a sequence of targets, and the backbone numerical algorithm is a plug-in. We chose ENCO+GIT as the main backbone because (i) GIT is a strong, widely used gradient-based targeting method, and (ii) ENCO is the standard backbone used in GIT, making comparisons more direct.
> To support generality, we have already: Paired LeGIT with CBED (Bayesian OED) and showed consistent improvements in SHD and SID on Alarm and Child (Table 3), demonstrating that LeGIT can be plugged into a non-gradient method and still provide gains.
>
> Follow your suggestion, we run `DCDI` on `Asia` dataset. Compared to `ENCO`, it requires significantly higher computational complexity [1]. Therefore, following the practice of GIT, we only tested it on the small graph asia (8 nodes). We set it to acquire 200 intervention data per round, for a total of 10 epochs, with only $T_{warmup}=2$ for LeGIT (DCDI all edges are present during the initialization of DCDI, there are no isolated nodes in the early stage). All other settings are the same as in the paper.
>
> | Method      | MeanSHD |
> |------------|---------|
> | RoundRoubin | 10.2   |
> | CBED       | 9.8     |
> | LeGIT+CBED | 8.4     |
> | GIT        | 8.6     |
> | LeGIT      | 8       |
>
> From the results, we can find that, LeGIT still posits advantages over previous methods with other active causal discovery algorithms as the backbone.

---

> ### Author Response · Authors · 2025-12-02
> **Response to Reviewer i43n (Part 2)**
>
> > W4.1: Discussion on the insufficient information about variables
>
> We agree this is an important point. LeGIT explicitly relies on textual meta-information for variables (Appendix E lists the descriptions used for each dataset). For example, Insurance  **One could find that the descriptions provided in Appendix E are relatively simple, while LLM can still provide effective guidance on early intervention targeting**. Therefore, we could infer that LLMs can still help without too much detailed description of the variable.
>
> If descriptions are missing or non-informative, the method effectively reduces to the numerical backbone alone. In this case, LLM’s response actually degenerates into a random selection. Nevertheless, **it also applies to humans**. So long as we have some prior understanding and knowledge about variables, our experiments demonstrate that the LLMs can provide effective guidance competitive to humans.
>
> We will include the aforementioned discussion in the limitation section of our manuscript.
>
>
>
> > W4.2: Discussion about the effectiveness of the proposed prompt and the consistency of LLMs' responses.
>
> We use self-consistency: for each dataset, we sample multiple LLM completions with shuffled variable order and aggregate to obtain the final warmup list. And we show robustness across different LLMs (GPT-4O, DeepSeek variants, Llama-3.1-405B) in Appendix C.3, where LeGIT still improves over GIT across backbones.
>
> > W5: scalability and cost-efficiency claim
>
> What we mean by “scalable and cost-efficient” is to compare with a human expert. In our human baseline, five graduate-level participants are asked to design interventions from the same descriptions; LeGIT achieves better or comparable performance while requiring only a small, fixed number of LLM calls that do not grow with the number of rounds.
>
> >Q1: Prompt for bootstrapped-warmup intervention set
>
>  We apologize that this was not clearly spelled out. The bootstrapped prompt is identical in structure to the warmup prompt templates in Appendix E, but applied only to the subset of “unvisited” variables identified from the intermediate graph after the first warmup rounds.
>
> >Q2: statistical significance
>
> We prioritized the main datasets in the text for brevity, but we included full statistical significance tests for all datasets in Appendix C.4 (Table 7). **The results show LeGIT is statistically significant across the board (e.g., p-values < 0.10 for all, mostly < 0.05)**.
>
> >Q3: example where rapid & costly interventions are required
>
> We realize that the phrase “rapid interventions are required” can be misleading, and we appreciate the opportunity to clarify it.
>
> We need to clarify that it does not mean that the physical act of intervening must be instantaneous, but rather that in many realistic online settings (e.g., biology experiments), interventions are expensive and one only has access to a **small number of intervention rounds**. Or alternatively, one wishes to have most outcomes to obtain an overview of the underlying causal relations, given a limited budget.
>
>
>
> >Q4: Typo
>
> Thank you for pointing this out. We will correct this typo and carefully proofread the final version for similar issues.
>
>
> Reference: Lippe P, Cohen T, Gavves E. Efficient neural causal discovery without acyclicity constraints[J]. arXiv preprint arXiv:2107.10483, 2021.

---

### Author Response · Authors · 2025-12-03
**Summary of Rebuttal**

Rebuttal Summary

We sincerely thank the reviewers and ACs for the generous time and patience in reviewing this work. Below, we provide a summary of key points raised by reviewers and how we address them.

To begin with, we summarize the key contributions of this work:
- We are the first to investigate the use of LLMs in experimental design and intervention targeting for causal discovery. The investigation is considered important by Reviewer i43n and `highly engaging` by Reviewer dDFN.
- We propose the first framework LeGIT that combines the strengths of LLMs and previous numerical methods GIT, which is considered novel and interesting by Reviewers tK3A, dDFN, AwPk.
- We demonstrate that LeGIT achieves state-of-the-art performance across multiple real-world benchmarks, compared to previous numerical-based methods as well as humans. The results are considered positive by Reviewer tK3A.
In addition, our presentations are widely appreciated by all reviewers.

Initially, we received one positive recommendation of scoring 8, and three negative recommendations of scores 2. Nevertheless, reviewers share similar concerns that we believe are addressable. Please find a summary of our responses below:

---

### Generality and applicability to other active causal discovery methods (i43n, dDFN, tK3A)

In the original manuscript, we provided the results of LeGIT with both ENCO and CBED methods to demonstrate that LeGIT is essentially backbone-agnostic. Following the suggestion by Reviewer i43n, we also extended the results to DCDI, where LeGIT also showed improvements over previous methods.

---

### Statistical significance of advantages of LeGIT (i43n, tK3A, AwPk)

While our main tables report results of 5 random seeds following the standard practice in the literature[1], we also provided results of 10 seeds for the Alarm and Child datasets to ensure statistical validity. As detailed in Appendix C.4, LeGIT consistently outperforms the baselines with 10 seeds. **We performed paired t-tests and Wilcoxon signed-rank tests, obtaining p-values well below 0.05, confirming that the improvements are statistically significant and not due to random variance**.

[1] Lippe P, Cohen T, Gavves E. Efficient neural causal discovery without acyclicity constraints[J]. arXiv preprint arXiv:2107.10483, 2021.

---

### Data contamination issue (tK3A, AwPk)

Observing the structure and generating accurate intervention targets are fundamentally different tasks. There is very limited documentation on how the causal graphs used in our experiments were originally obtained via interventional experiments, so it is unlikely that LLMs have been trained on, or can directly memorize, the intervention selection process itself.

**Importantly, Jiralerspong et al. (2024) and Khatibi et al. (2024) use GPT-4–class models to directly predict edges from variable descriptions on exactly the kind of benchmarks we study**. These works show that even with access to rich metadata, GPT-4–level models do not recover the correct structures for several datasets we use (e.g., Alarm/Child), even if they may have seen related content during pretraining. This suggests that current LLMs are not simply memorizing and reproducing the ground-truth graphs.

To further address the concern that GPT-4o might just “know” these specific datasets, we also evaluated LeGIT with open-source models such as Llama-3.1-405B and DeepSeek-V3 (Appendix C.3). **These models are trained on different data and likely have different contamination profiles, yet LeGIT consistently maintained its performance advantage across them**. This cross-model robustness makes a “memorized benchmark” explanation less plausible.

---

### Discussion on the use of meta-information (i43n, dDFN)

The focus of this work is to investigate whether and how LLMs can utilize the textual meta-information to facilitate the active causal discovery process, which is widely available in the real world.

We show that relatively simple descriptions of variables can already help accelerate the active causal discovery. (Appendix E lists the descriptions used for each dataset). For example, Insurance  **One could find that the descriptions provided in Appendix E are relatively simple, while LLM can still provide effective guidance on early intervention targeting**.

When those descriptions are missing or non-informative, the method effectively reduces to the numerical backbone alone. In this case, LLM’s response actually degenerates into a random selection. Nevertheless, **it also applies to humans**. So long as we have some prior understanding and knowledge about variables, our experiments demonstrate that the LLMs can provide effective guidance competitive to humans.

---

### Remaining concerns

We also provided clarifications, such as algorithm and prompt design, the selection of datasets, point-by-point, to the potentially misunderstood and confusing points raised by reviewers.

---

### Meta-Review · Area_Chair_C2nW · 2026-01-04

**Summary:**

The reviewers are concerned about out-of-distribution generalization while using LLM on an unknown variable, and unfair performance comparison with the baselines.

**Reviewer Concerns:**

Reviewer i43n brought up Eberhardt's work. Indeed there is a long line of experimental/intervention design papers none of which are cited by the authors. Combined with Reviewer dDFN's concerns about the positioning of the paper and unfairness towards the baselines this becomes an important issue for a paper claiming a contribution in the space of experimental design.

I also do not think Reviewer dDFN would be fully convinced to switch to acceptance territory although they may have slightly increased their score. This is because we really do not have a reason to expect LLM to outperform specialized baselines if LLM is not trained on the side information that the baseline algorithms aren't designed to use.

**Reviewer Scores:**

I do not believe Reviewer i43n's score would go up given that their concern is about offloading a lot of the work to LLM. The reviewer questions what if the LLM does not know about the variables, has not seen it in its training data, a fair question. There is really no way to answer this question and it connects to the question of whether LLMs are indeed performing causal inference or performing look up. I do not believe the authors had a satisfactory answer to this.

---

### Decision · Program_Chairs · 2026-01-26

Reject